# High expression of Tie-2 predicts poor prognosis in primary high grade serous ovarian cancer

Minna Sopo[1]*, Hanna Sallinen[1,2,3,4], Kirsi Hämäläinen[5,6], Annukka Kivelä[2,4], Seppo Ylä-Herttuala[2,4], Veli-Matti Kosma[5,6,7], Leea Keski-Nisula[1,3], Maarit Anttila[1,3]

1 Department of Gynecology, Kuopio University Hospital, Kuopio, Finland, 2 Department of Biotechnology and Molecular Medicine, A.I.Virtanen Institute for Molecular Sciences, University of Eastern Finland, Kuopio, Finland, 3 Institute of Clinical Medicine, School of Medicine, Gynaecology, University of Eastern Finland, Kuopio, Finland, 4 Gene Therapy Unit, Kuopio University Hospital, Kuopio, Finland, 5 Department of Pathology and Forensic Medicine, Kuopio University Hospital, Kuopio, Finland, 6 Pathology and Forensic Medicine, University of Eastern Finland, Kuopio, Finland, 7 Cancer Center of Eastern Finland, University of Eastern Finland, Kuopio, Finland

* Minna.Sopo@kuh.fi

**Data Availability Statement:** All relevant data are within the manuscript and its Supporting Information files.

## Abstract

### Background

Antiangiogenic therapy, although part of standard treatment in ovarian cancer, has variable efficacy. Furthermore, little is known about the prognostic biomarkers and factors influencing angiogenesis in cancer tissue. We evaluated the expression of angiopoietin-2 and two endothelial tyrosine kinase receptors, Tie-1 and Tie-2, and assessed their value in the prediction of survival in patients with malignant epithelial ovarian cancer. We also compared the expression of these factors between primary high grade serous tumors and their distant metastasis.

### Materials and methods

We evaluated 86 women with primary epithelial ovarian cancer. Matched distal omental metastasis were investigated in 18.6% cases (N = 16). The expression levels of angiogenic factors were evaluated by immunohistochemistry in 306 specimens and by qRT-PCR in 111 samples.

### Results

A high epithelial expression level of Tie-2 is a significant prognostic factor in primary high grade serous ovarian cancer. It predicted significantly shorter overall survival both in univariate (p<0.001) and multivariate survival analyses (p = 0.022). Low angiopoietin-2 expression levels in primary ovarian tumors were significantly associated with shorter overall survival (p = 0.015) in the univariate survival analysis. A low expression of angiopoietin-2 was also significantly related to high grade tumors, size of residual tumor after primary surgery and the recurrence of cancer (p = 0.008; p = 0.012; p = 0.018) in the whole study population. The expression of angiopoietin-2 and Tie-2 was stronger in distal omental metastasis than in primary high grade serous tumors in matched-pair analysis (p = 0.001; p = 0.002).

**Funding:** This work was supported by: MS: The Emil Aaltonen Foundation, MS: The Cancer Foundation of the Northern Savo, MS: The Kuopio University Hospital Research Foundation, HS: The Finnish Medical Foundation and Kuopio University VTR grant. The funders had no role in study design, data collection and analysis, decision to publish, or preparation of the manuscript.

**Competing interests:** The authors have declared that no competing interests exist.

## Conclusions

The angiogenic factor, angiopoietin-2, and its receptor Tie-2 seem to be significant prognostic factors in primary epithelial ovarian cancer. Their expression levels are also increased in metastatic lesions in comparison with primary tumors.

## Introduction

The formation of new blood vessels, i.e. angiogenesis, is essential for the growth and dissemination of malignant tumors. There are several growth factor pathways involved in regulating and maintaining the angiogenic switch to promote carcinogenesis. For example, the vascular endothelial growth factor (VEGF) family has been well characterized and is considered to be the most important inducer of angiogenesis and lymphangiogenesis including factors such as VEGF-A, -C, -D and placental growth factor (PLGF) [1, 2].

Other endothelial growth factor systems, such as angiopoietin-Tie complex, have not been as extensively studied. Angiopoietin-2 (Ang-2) is produced by endothelial cells and it signals through two tyrosine kinase receptors, Tie-1 and Tie-2. Both angiopoietin-1 (Ang-1) and Ang-2 are required for the vascular remodeling and maturation. Ang-2 functions as an autocrine controller of endothelial cells in a context-dependent manner promoting either vessel growth or regression depending on the levels of other growth factors, such as VEGF-A. Tie-1 and Ang-2 are also essential for lymphatic development [1–3].

In more than 70% of cases, ovarian cancer has metastasized by the time of diagnosis, which makes the 5-year prognosis very low, approximately 19–28% in stages III and IV [4]. There is no curative treatment for the disseminated or recurrent cancer, despite the high initial chemosensitivity to platinum-based chemotherapy [5]. As our knowledge of the molecular mechanisms of cancer biology has expanded, this has accelerated the development of targeted, individualized treatment strategies. Antiangiogenic treatment with a humanized monoclonal VEGF-antibody, bevacizumab, has become standard care in primary and recurrent ovarian cancers and several multi-target tyrosine kinase inhibitors of VEGF-pathway have been introduced into the treatment of other types of cancer [6–8]. However, some problems have emerged with these new therapies which have prevented the more widespread use of this kind of therapy e.g. the development of resistance deteriorating clinical outcomes and severe side-effects such as hypertension, proteinuria, wound-healing complications, thrombotic events and gastrointestinal perforations. Trebananib, a peptide-Fc fusion protein, which binds both Ang-1 and Ang-2 preventing their interaction with Tie-2 receptor, has prolonged median progression free survival (PFS) by two months in phase 3 studies and showed antitumor activity with less class-specific adverse events in patients with recurrent ovarian cancer [9].

The development of targeted treatments will demand more sophisticated patient selection by exploitation of tissue and plasma/serum biomarkers; these will open new avenues in the prediction of clinical prognosis and the response to therapy [10]. It has been reported that the levels of Ang-1 and Ang-2 have been elevated in preoperative serum samples of women with ovarian cancers as compared to the samples of healthy women or women with benign or borderline ovarian neoplasms [11]. However, there are only four studies with smaller populations and none investigating the tissue expressions of Ang-2 and both of its receptors in ovarian cancer (S1 Table in S1 File) [12–15]. The role of Ang-2 expression in tumor tissue and its possible association with the mechanisms of malignant angiogenesis and further on dissemination and prognosis are still unclear.

In this study, we performed immunohistochemical evaluation of Ang-2, Tie-1 and Tie-2 receptors in primary ovarian epithelial cancer tissue samples of 86 women and further in 16 related metastatic tumors. In addition, we measured the expression of these angiogenic factors using qRT-PCR in 22 primary and 15 metastatic tumors. Our principal aims were (1) to determine whether the expression predicted the clinical course, prognosis and survival of women with epithelial ovarian cancers and (2) to compare the expression levels between primary tumors and their related metastases. As far as we are aware, this is the first study to evaluate Ang-2 and its receptors using both immunohistochemistry and qRT-PCR in primary ovarian cancer and related metastatic tumors from the same patients, and to correlate the expression data with the clinical outcome.

## Materials and methods

### Patients and data collection

This study investigated a total of 86 women who were diagnosed with malignant ovarian epithelial cancers in Kuopio University Hospital (KUH) between 1.9.1999 and 28.3.2007. Table 1 summarizes the characteristics of the studied women. Tumor tissue samples, obtained from treatment naïve patients, and patient information were prospectively collected, and retrospectively analyzed using immunohistochemistry and qRT-PCR. The follow-up time ended in March 2017. In a subgroup of 16 patients with high grade serous ovarian carcinoma, we also analyzed related omental metastases at the time of primary diagnosis. Histological type and grade were evaluated according to the World Health Organization (WHO) Classification of Tumors [16]. Women with nonepithelial neoplasms, those treated prior to the operation, and those who were not operated at all, were excluded from the study.

Epithelial ovarian cancers were operatively staged according to the criteria of the International Federation of Gynecology and Obstetrics (FIGO). Most of the included women were treated with platinum-based chemotherapy after the operation. A significant proportion, 42%, of the patients (36 patients) received paclitaxel and carboplatin as their primary adjuvant treatment; 30% (26 patients) started with paclitaxel and carboplatin, but finished with carboplatin on its own, because of side effects. Fewer, 13% (11 patients), were only treated with carboplatin, 3% (three patients) were treated with cisplatin combined with cyclophosphamide, 2% (two patients) with gemsitabine and carboplatin and 1% (one patient) with cisplatin and paclitaxel. Seven patients either did not receive chemotherapy because of the stage IA cancer or the information of the treatment was not available.

Four patients were treated with bevacizumab in their recurrent phase of the disease, and therefore none of the patients had received it as part of their first line treatment. All of them had ST III-IV high grade serous carcinoma (HGSC) except for one who had a transitional cellular histology. They were diagnosed at the age of 45–59 years and two of them enjoyed a complete response after bevacizumab, while in the remaining two women, the cancer progressed. Seven of the patients included in the study had been tested for the presence of the BRCA mutation. Two patients had germline BRCA1 and two had a germline BRCA2 mutation. One patient had a somatic BRCA2 mutation. Written informed consent was obtained from the patients participating in this study. The study was approved by the Ethical Committee of the KUH (26/99) [17].

### Immunohistochemistry

Tissue samples were embedded in paraffin and cut into 5-μm-thick sections. Next, the sections were processed for staining with hematoxylin-eosin, monoclonal Ang-2 antibody (mouse, code NB110-85467, dilution 1:250, Novus, ID AB 1199462), polyclonal Tie-1 (rabbit, code

**Table 1. Clinical characteristics of the ovarian cancer patients.** All patients are white females (Finnish, Caucasian).

| Characteristic | Ovarian carcinoma N (%) |
|---|---|
| **Total** | 86 (100) |
| **Median age at diagnosis, years** | 58 (26–88) |
| **Histologic subtype** | |
| Serous | 51 (59) |
| High grade | 45 |
| Low grade | 6 |
| Mucinous | 11 (13) |
| Endometrioid | 15 (17) |
| High grade | 13 |
| Low grade | 2 |
| Clear cell | 5 (6) |
| Other | 4 (5) |
| **Ca12-5, median, kU/l** | 363 (5–10100) |
| **FIGO Stage** | |
| I | 12 (13) |
| II | 10 (12) |
| III | 46 (53) |
| IV | 18 (22) |
| **Histological grade**[*] | |
| 1 | 9 (12) |
| 2 | 25 (33) |
| 3 | 41 (55) |
| **Ascites** | 60 (70) |
| **No ascites** | 16 (19) |
| **No data on ascites** | 10 (11) |
| **Residual tumor at primary surgery** | |
| None | 40 (47) |
| $\leq$1 cm | 7 (8) |
| >1 cm | 39 (45) |
| **Chemotherapy response** | |
| Complete response | 57 (66) |
| Partial response | 4 (5) |
| Stable disease | 2 (2) |
| Progressive disease | 5 (6) |
| No chemotherapy | 5 (6) |
| No data | 13 (15) |
| **Tumor recurrence** | 48 (56) |
| **No recurrence** | 25 (29) |
| **No data on recurrence** | 13 (15) |
| **Patient status** | |
| Dead from ovarian cancer | 52 (60) |
| Alive | 29 (33) |
| **Median follow-up, months** | 65 (0–198) |

Values are presented as *n* (%) or as indicated units (range).

[*]Mucinous tumors are not graded.

ab64477, dilution 1:200, Abcam, ID1143463), polyclonal Tie-2 (rabbit, code sc-9026, dilution 1:50, Santa Cruz, ID2203226) [18–20]. 30 human tissue samples were included as positive and negative controls in this study: 22 positive and eight negative controls. Expression of the antibodies was equally detected in the cytoplasm of the tumor epithelial cells, vascular endothelium and stroma in the positive controls (including adnexal, uterine and tonsillar tissue) as in positive evaluated samples. Samples without primary antibody were used as negative controls (S1 Fig).

Immunostaining of Ang-2, Tie-1 and Tie-2 was microscopically evaluated by M.S. and K.H (Leitz Wetzlar 512 761/20; Germany). During the evaluation of immunostained sections, the investigators were blinded to the patients' clinical situation. After screening the whole section, ten randomly selected microscopic fields were examined at ×200 magnification (×20 objective lens and ×10 ocular lens; 0.74 mm$^2$ per field) to determine the mean percentage of specific immunostained epithelial tumor cells. The percentage of positively stained cells (PP) in the sample was assigned as a numerical score: 0, negative; 1, <10%; 2, 11–50%; 3, 51–80%; and 4, >80% positive cells. The intensity (SI) of the immunostained areas was defined generally as follows: 0, negative; 1, weak; 2, moderate; 3, strong. An immunoreactive score (IRS) ranging from 0–12 was calculated using the following formula: IRS = PP × SI [17, 21]. IRS and PP values were divided to high and low groups according to the median value of each group: low Ang-2 IRS ≤ 6, PP ≤65, high Ang-2 IRS >6, PP >65 low Tie-1 IRS ≤2, PP ≤32,5, high Tie-1 IRS >2, PP >32,5, low Tie-2 IRS ≤2, PP ≤35, high Tie-2 IRS >2, PP >35.

Stromal immunostaining was evaluated by ranking the staining intensity: 0, negative; 1, weak; 2, strong. However stromal expression was positive in all samples, and thus the staining intensity was further classified simply as weak or strong [17].

Endothelial immunostaining expression was characterized as either positive or negative. We also evaluated the cellular distribution of the staining (cytoplasm, nucleus, and cell membrane) in epithelial ovarian tumors [17].

## Quantitative real-time polymerase chain reaction (qRT-PCR) analysis

Samples from 22 primary and 15 metastatic tumors were analysed by qRT-PCR.

RNA was isolated using the TRI-reagent (Sigma Aldrich, St. Louis, Missouri, USA), and treated with DNase (Promega, Fitchburg, Wisconsin, USA). From 5 μg total RNA, cDNA was synthesized using random hexamer primers (Promega) and RevertAid™ reverse transcriptase (Fermentas, Waltham, Massachusetts, USA). We measured the relative expressions of the mRNAs encoding Ang-2, Tie-1 and Tie-2 by qRT-PCR using specific Assays-on-Demand target mixes (Applied Biosystems, Foster City, California, USA), following the manufacturer's protocol (StepOnePlus, Applied Biosystems). The expression levels were normalized to peptidylprolyl isomerase A [17, 22].

## Statistical analysis

Statistical analyses were performed using SPSS for Windows (version 24, 1989–2016, SPSS Inc., Chicago, USA). We performed the Kruskal-Wallis test followed by Mann-Whitney tests with continuous variables applying multiple comparisons when appropriate. For analyses of clinicopathological associations and survival analyses, PP and IRS values, and qRT-PCR levels were each dichotomized into low and high groups using the median values as a cut-off [2]. We used the chi-squared test to analyze frequency tables, and the Wilcoxon signed-rank test to compare histology and qRT-PCR results between primary ovarian tumors and the related metastases. Univariate survival analyses were based on the Kaplan-Meier method, and survival curves were compared using the log-rank test. Variables that were significant in the univariate

analyses were entered in a stepwise manner into the Cox regression multivariate analysis. Overall survival (OS) was defined as the time interval between the date of surgery and the date of death or the end of follow-up. Progression free survival (PFS) was defined as the time interval between the date of surgery and the date of identified recurrence or death. Correlations between histological parameters and qRT-PCR levels were analyzed using the Spearman's correlation test. A *p* value of <0.05 was considered significant [2, 17].

## Results

### Immunohistochemical analyses of Angiopoietin-2, Tie-1 and Tie-2 in primary ovarian tumors and in metastatic lesions

Ang-2 was mainly detected in the cytoplasm of tumor epithelial cells and in the vascular endothelium of primary tumors. Ang-2 was found to be the strongest factor according to the expression of all analyzed three factors in the epithelial cells of primary ovarian cancer (IRS 6) (Figs 1 and 2). On the contrary, the stromal staining of Ang-2 was mostly weak in 67% of cases in primary ovarian tumors (Table 2). In distal omental metastasis, the expression of Ang-2 was even stronger (IRS 9) with moderate or strong epithelial staining intensity in all of the samples and also strong stromal expression was detected in 75% of cases (Figs 1, 2A and 2B, Table 2).

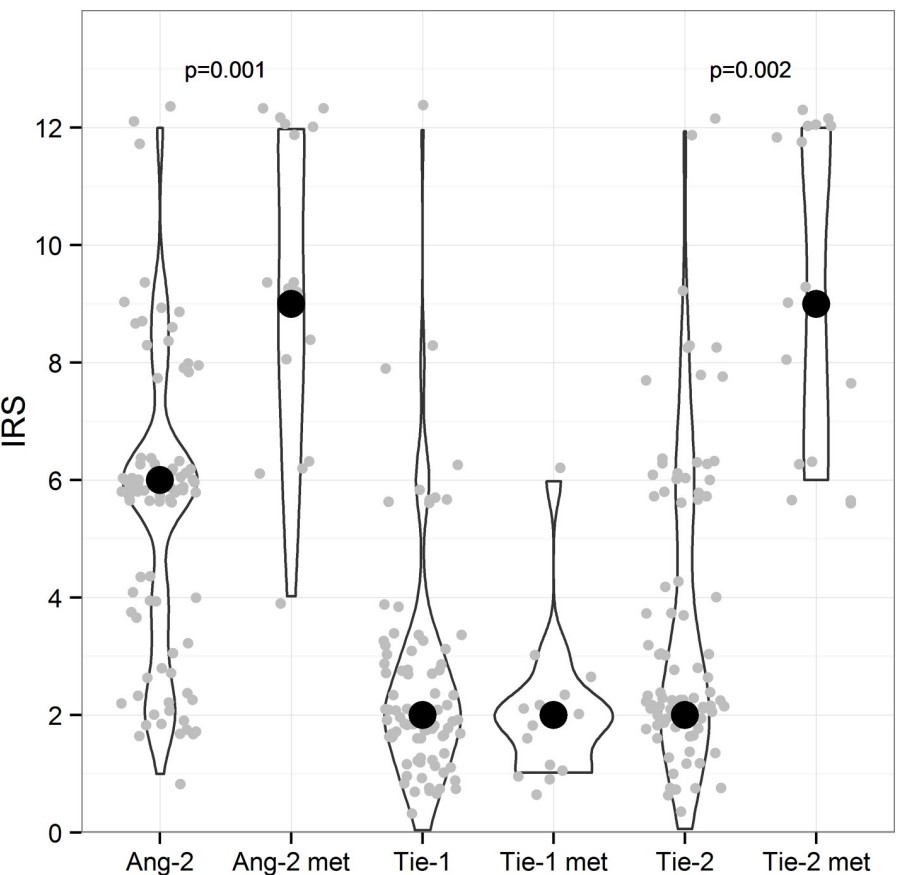

**Fig 1. Immunohistochemical staining scores of primary tumors and metastases.** IRS was calculated from the tumor epithelium of 86 primary ovarian tumors and 16 metastatic lesions (grey dots). Ang-2, Tie-1 and Tie-2 groups include 86 ovarian cancer patients each and Ang-2met, Tie-1met and Tie-2met groups have 16 patients each. Wilcoxon signed-rank test was used to compare the IRS values for primary tumors and related metastases. Significant p-values are marked in the Figure. The median IRS of the group is signed with a black dot. Ang-2 met Angiopoietin-2 metastasis, IRS immunoreactive score.

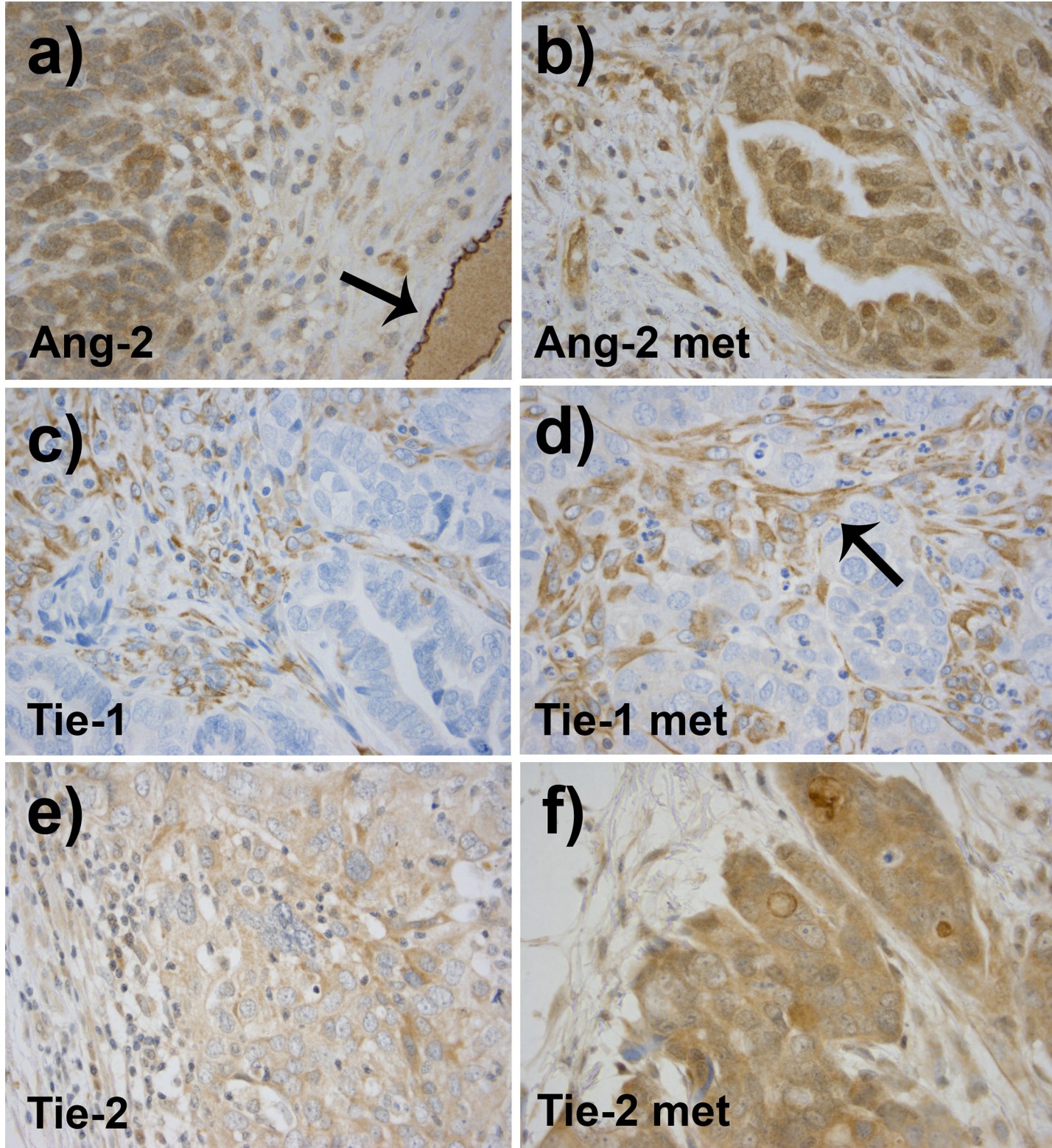

**Fig 2.** As compared to the primary tumors, related omental metastases showed stronger expression of Ang-2 (**a** and **b**) and Tie-2 (**e** and **f**). Primary tumors are shown on the left column and related metastases on the right. The vascular endothelial expression was strong in Ang-2 (**black arrow in a**). Tie-1 expression did not differ significantly between the primary and the distal metastatic tumors (**c** and **d**). Tie-1 was expressed in the tumor stroma (**black arrow in d**).

**Table 2. Expressions of Ang-2 and its receptors Tie-1 and Tie-2 in primary ovarian cancer and 16 high grade serous tumors with their related metastases.**

| Variable | Primary tumors | Primary HGSC | Related metastases | *p-value* |
|---|---|---|---|---|
| | (N = 86) | (N = 16) | HGSC (N = 16) | |
| **Ang-2** | | | | |
| IRS (0–12) | 6 (1–12) | 6 (2–6) | 9 (4–12) | **0.001** |
| PP % | 65 (10–90) | 62.5 (25–75) | 82.5(40–98) | **0.001** |
| SI | | | | |
| ≤Weak | 21 (24%) | 5 (31%) | 0 (0%) | **0.036***  |
| Moderate | 54 (63%) | 9 (56%) | 6 (37.5%) | |
| Strong | 11 (13%) | 2 (13%) | 10 (62.5%) | |
| **Stroma SI** | | | | |
| Weak | 58 (67%) | 10 (63%) | 4 (25%) | 0.145 |
| Strong | 28 (33%) | 6 (37%) | 12 (75%) | |
| **Tie-1** | | | | |
| IRS (0–12) | 2 (0–12) | 2 (1–6) | 2 (1–6) | 1.000 |
| PP % | 32.5(0–90) | 27.5 (5–70) | 35 (5–70) | 0.789 |
| SI | | | | |
| ≤Weak | 74 (86%) | 14 (88%) | 15 (94%) | 0.773 |
| Moderate | 11 (13%) | 2 (12%) | 1 (6%) | |
| Strong | 1 (1%) | 0 (0%) | 0 (0%) | |
| **Stroma SI** | | | | |
| Weak | 37 (43%) | 6 (37%) | 2 (12.5%) | 0.164 |
| Strong | 49 (57%) | 10 (63%) | 14 (87.5%) | |
| **Tie-2** | | | | |
| IRS (0–12) | 2 (0–12) | 2 (2–8) | 9 (6–12) | **0.002** |
| PP % | 35 (0–90) | 30 (15–85) | 85 (65–98) | **0.001** |
| SI | | | | |
| ≤Weak | 54 (64%) | 11 (69%) | 0 (0%) | 0.803 |
| Moderate | 28 (33%) | 5 (31%) | 7 (44%) | |
| Strong | 3 (3%) | 0 (0%) | 9 (56%) | |
| **Stroma SI** | | | | |
| Weak | 50 (59%) | 6 (37%) | 0 (0%) | no statistics** |
| Strong | 35 (41%) | 10 (63%) | 16 (100%) | |

IRS and PP are presented as median values (range), SI is presented as N (%). IRS immunoreactive score, PP percentage of positively stained epithelial cells, SI staining intensity, HGSC high grade serous cancer. P-values are estimated by Wilcoxon signed-rank test for IRS, PP and Pearson's chi-square test for SI of the primary and related metastatic tumors.

* 3 cells (75%) have expected count less than 5.

**SI of stroma in metastatic lesions is constant.

Tie-1 expression was the weakest of all evaluated factors in the epithelial cells of primary and metastatic tumors (IRS 2) (Figs 1, 2C and 2D, Table 2). The staining intensity was negative or weak in 86% of all samples of the primary ovarian cancer. In contrast, stromal staining was strong in 57% of cases, emphasizing stromal fibroblasts and vascular endothelium (Table 2).

Tie-2 was expressed mainly moderately or weakly in the cytoplasm of epithelial cells in the primary cancer (IRS 2) (Figs 1 and 2E). In most samples, the cell membranous staining was evident and it was particularly strong in some cases. Stromal expression was strong in 41% of the primary ovarian tumors and the endothelial staining was positive in all samples (Table 2). With respect to Tie-2 staining, the greatest contrast was exhibited between expression in the

primary and the metastatic tumor. All metastatic samples displayed moderate or strong epithelial cytoplasmic expression of Tie-2 (IRS 9) and strong stromal staining (Figs 1, 2E and 2F). The results of the immunohistochemical analysis of the whole study population is presented in Table 2.

The level of Tie-1 expression was moderately, but significantly, correlated with its ligand Ang-2 IRS (r = 0.5, p < 0.001) in primary serous ovarian tumors, and the correlation was also significant with other histological types (IRS r = 0.3, p = 0.001; PP r = 0.4, p <0.001). The expression level of Tie-2 showed a weak, but significant correlation with Ang-2 PP (r = 0.3, p = 0.012) and moderate correlation with Tie-1 IRS and PP (r = 0.5, p <0.001; r = 0.4, p <0.001) in primary cancer of the whole studied population. In high grade serous group the Ang-2 PP did not correlate with the Tie-2 PP. Correlations were tested with the parameters of tumor epithelial expression, IRS and PP using the Spearman's test.

## Expression of angiogenic factors in primary high grade serous tumors as compared to related metastases

Expression of both Ang-2 and Tie-2 in high grade serous tumors was significantly stronger in omental metastases than in their related primary tumors. Metastases showed a significantly higher median IRS of Ang-2 (9, mean 9.1, 95% CI 7.7–10.6) than primary tumors (6, mean 5.5, 95% CI 4.5–6.0) (p = 0.001) (Figs 1 and 2, Table 2).

In the metastases, Tie-2 IRS was significantly higher compared to those of primary tumors, IRS 9.25 (95% CI 7.8–10.7) vs. 3.55 (95% CI 3.0–4.1) (p = 0.002). The level of Tie-1 epithelial expression did not differ between primary and related metastatic lesions (Figs 1 and 2, Table 2).

## Correlation of qRT-PCR results with immunohistochemical staining

Ang-2 IRS was significantly negatively correlated to the mRNA levels of Ang-2 measured by qRT-PCR in primary ovarian cancer (r = -0.64, p = 0.002). The correlation was even stronger when only high grade serous tumors were counted (r = -0.868, p < 0.001) (S2 Table in S1 File). In metastatic tumors Ang-2 PP was strongly correlated to the corresponding mRNA levels (r = 0.752, p = 0.005) (S2 Table in S1 File). Tie-1 and Tie-2 qRT-PCR values correlated strongly to each other (r = 0.919, p < 0.001), but did not have statistically significant correlations to the respective immunohistochemical expression. In metastases, mRNA-levels of Ang-2 correlated strongly with Tie-2 mRNA (r = 0.61, p = 0.016).

## Reltion of clinicopathological data to the expression of Ang-2, Tie-1 and Tie-2 receptors

The low epithelial expression of Ang-2 in primary tumors associated significantly with ovarian cancer recurrence and with greater residual tumor size (>1cm) after primary surgery (p = 0.018, p = 0.012) (S3 Table in S1 File). When including only serous primary tumors in the analysis, the epithelial expressions of Ang-2 were significantly lower in stages III–IV than in stages I–II (p = 0.047).

(S3 Table in S1 File). In high grade serous tumors, high levels of Ang-2 staining were related to resistance to platinum chemotherapy (p = 0.017).

In women with primary serous ovarian malignancies, low epithelial expression levels of Tie-1 were significantly associated with a recurrence of ovarian cancer (p = 0.025), with a greater residual tumor size (>1cm) after primary surgery (p = 0.008) and with high grade tumors (p = 0.006) (S3 Table in S1 File). Tie-1 expression was also related to the histological subtype. Mucinous tumors displayed stronger Tie-1 expression than serous tumors (p = 0.029)

and clear cell lesions were associated with a lower Tie-1 expression (p = 0.001) (S3 Table in S1 File). When all histologic types were taken into account, high grade tumors were associated with low Tie-1 expression (p = 0.018) (S3 Table in S1 File). Strong stromal Tie-1 intensity was related to the stage III–IV (p = 0.038) in high grade tumors.

Tie-2 expression was stronger in mucinous tumors than in serous cancer (p = 0.001) and it was lower in epithelial cells of clear cell histology (p = 0.024) (S3 Table in S1 File).

Other clinical parameters were not significantly associated with the immunohistochemical data. We obtained equivalent results when analyzing the associations with Pearson's chi-squared test and as continuous parameters with the Kruskal-Wallis test (S3 Table in S1 File).

## Overall survival and progression-free survival of ovarian cancer patients

The median follow-up time was 65 months (range, 0–198 months). At the end of follow-up, 57 patients (66%) had died. The median OS was 77 months (95% CI, 59–95 months) and the 5-year survival rate was 57% (95% CI, 46–68%).

High epithelial expression of **Tie-2 was an independent significant prognostic factor in primary high grade serous** ovarian cancer (Table 3). The 5-year survival rate of the patients with high Tie-2 expression was only 29% in contrast to patients with low Tie-2 expression, whose 5-year survival was 54% (Table 4). High Tie-2 predicted a shorter overall survival both in univariate and multivariate survival analysis (p <0.001, univariate; p = 0.022, multivariate) in HGSC (Fig 3). The significance persisted also when high grade endometrioid tumors were taken into account (p = 0.028). Furthermore, low Ang-2 expression levels in primary ovarian tumors predicted poor overall survival (OS) in the univariate analysis (p = 0.015) of the whole study population (Fig 3). If we examined all of serous tumors, a shorter OS was associated with a high expression of Tie-2 receptor (p = 0.012) (Table 3).

Advanced stage, the presence of residual primary tumor, incomplete primary response to chemotherapy, and the presence of ascites predicted poor OS in the univariate survival analysis. In the multivariate analysis, along with Tie-2, OS remained significantly associated with the presence of residual primary tumor (p <0.001) and incomplete response to chemotherapy (p <0.001) (Fig 3, Tables 3 and 4).

PFS analysis included a total of 67 patients, of whom 48 (71%) experienced recurrence during the follow-up. The median PFS was 15 months (95% CI, 9–21 months).

In both the univariate and multivariate analyses, advanced stage (p = 0.022 in multivariate analysis) and the presence of ascites (p = 0.045 in multivariate analysis) predicted a significantly shorter PFS time (Table 3).

## Discussion

This study is the first to date, which has analysed both Ang-2 and the protein levels of its receptors Tie-1 and Tie-2 in the same patient population with epithelial ovarian cancer at the tissue level using both immunohistochemistry and qRT-PCR in primary tumors and related distal omental metastases and correlating the results to the clinical outcome of the patients. We found that a high epithelial expression of Tie-2 in primary high grade serous ovarian cancer predicted a shorter OS both in univariate and multivariate survival analyses. Furthermore, a low epithelial expression of Ang-2 in primary ovarian tumor cells predicted poor overall survival. A low epithelial level of Ang-2 expression was also associated with the recurrence of cancer, with a larger residual tumor after the surgery and with a higher stage. In addition, the expressions of Tie-2 and Ang-2 were significantly higher in related distal omental metastases of high grade serous cancer than in primary tumors, highlighting the importance of angiogenic factors in the metastatic process of ovarian cancer.

**Table 3. Overall survival and progression-free survival according to immunohistochemical staining of Ang-2, Tie-1, Tie-2 and clinical characteristics.**

| Variable | Univariate analysis | Multivariate analysis<br>Hazard ratio, 95% CI | | *p* |
|---|---|---|---|---|
| **Overall survival** | | | | |
| **Immunohistochemical staining** | | | | |
| Ang-2 IRS | **0.015** | 2.58 | 1.17–5.72 | 0.308 |
| Ang-2 IRS (HGSC) | 0.628 | 0.77 | 0.27–2.20 | 0.631 |
| Tie-1 IRS | 0.767 | 1.09 | 0.62–1.90 | 0.768 |
| Tie-2 IRS | 0.409 | 1.24 | 0.74–2.09 | 0.412 |
| Tie-2 IRS (serous) | **0.012** | 2.36 | 1.14–4.55 | 0.117 |
| Tie-2 IRS (HGSC) | **<0.001** | 2.75 | 1.16–6.54 | **0.022** |
| **Clinical features** | | | | |
| Stage | **<0.001** | 10.01 | 2.38–42.06 | 0.095 |
| Ascites | **0.046** | 3.11 | 1.202–8.05 | 0.106 |
| Primary residual tumor | | | | |
| None | | | | |
| <1cm | **<0.001** | 4.21 | 1.83–9.66 | **0.001** |
| >1cm | **<0.001** | 4.39 | 2.16–8.94 | **0.001** |
| Chemotherapy response | | | | |
| Complete response | | | | |
| Partial response | **<0.001** | 3.75 | 1.32–10.67 | **0.013** |
| Stable disease | **<0.001** | 23.77 | 4.33–130.45 | **<0.001** |
| Progressive disease | **<0.001** | 12.90 | 4.67–35.61 | **<0.001** |
| **Progression-free survival** | | | | |
| **Immunohistochemical staining** | | | | |
| Ang-2 IRS | 0.364 | 1.53 | 0.59–3.99 | 0.383 |
| Tie-1 IRS | 0.445 | 1.33 | 0.62–2.81 | 0.464 |
| Tie-2 IRS | 0.860 | 1.06 | 0.53–2.15 | 0.865 |
| Tie-2 IRS (HGSC) | 0.073 | 2.09 | 0.87–5.02 | 0.100 |
| **Clinical features** | | | | |
| Stage | **<0.001** | 45.09 | 2.74–742.25 | **0.022** |
| Ascites | **0.028** | 2.15 | 1.02–4.55 | **0.045** |

Ang-2 angiopoietin-2, IRS immunoreactive score, HGSC high grade serous carcinoma.

**Table 4. Overall survival (%) and hazard ratios (HR) of angiopoietin-2, Tie-1 and Tie-2 expressions of ovarian cancer patients.**

| | 5-year survival<br>% | Cox univariate<br>HR | CI 95% | p |
|---|---|---|---|---|
| **Ang-2 IRS** | | | | |
| low (≤6) | 48 | 2.59 | 1.17–5.72 | **0.019** |
| high (>6) | 82 | | | |
| **Tie-1 IRS** | | | | |
| low (≤2) | 53 | 1.09 | 0.62–1.90 | 0.768 |
| high (>2) | 62 | | | |
| **Tie-2 IRS** | | | | |
| **(HGSC)** | | | | |
| low (≤2) | 54 | 3.44 | 1.67–7.10 | **0.001** |
| high (>2) | 29 | | | |

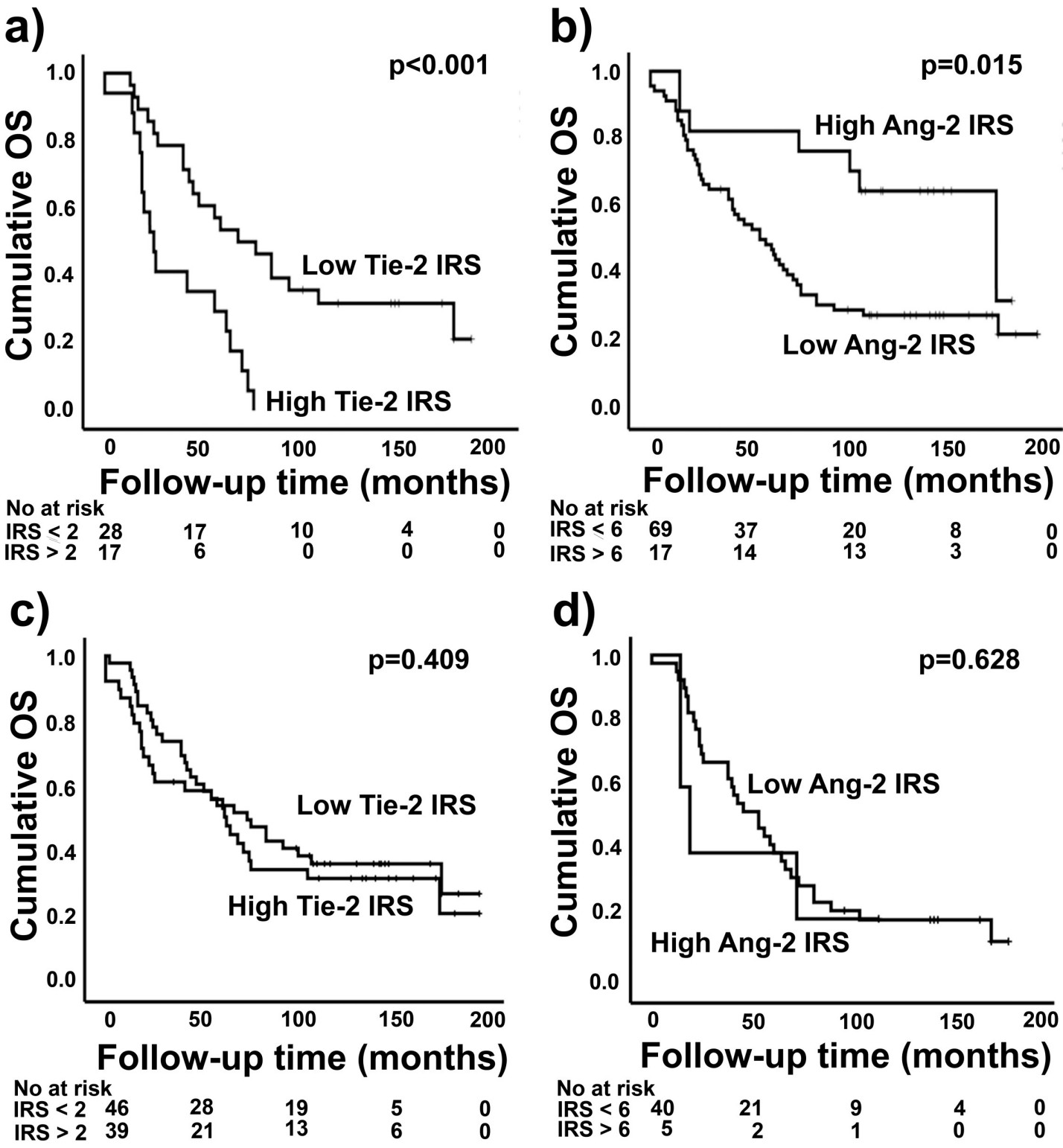

**Fig 3. Kaplan Maier curves of the immunohistochemical biomarkers as prognostic factors in ovarian cancer patients.** Poor overall survival (OS) was significantly associated with high Tie-2 immunoreactive score (IRS) (**a**) in the univariate and multivariate analyses of primary high grade serous tumors. Short OS was also associated with low angiopoietin-2 (Ang-2) IRS of all ovarian cancer patients (**b**). Tie-2 IRS of all histologies and grades together (**c**) and Ang-2 IRS of only high grade serous tumors (**d**) did not reach significance.

Previously, Ang-2 was thought to be primarily expressed in stromal endothelial cells and detectable in tumor cells in only a small percentage (12%) of tumor specimens collected from human ovarian cancer patients [12]. On the contrary, it was found here that tumor cells commonly widely expressed Ang-2 with moderate to strong intensity, which is in line with the report of Brunckhorst et al. [15]. Tumor epithelial expression of Ang-2 has also been detected in gastric, bladder, breast, hepatocellular, colorectal and salivary gland cancers [23–26]. In our study, the most aggressive tumors expressed less Ang-2 in tumor epithelial cells which is unexpected since previously, Ang-2 has not correlated to the clinicopathological factors [13] or overexpression has even been associated with an aggressive phenotype [24].

We also found that the tumor endothelial expression of Ang-2 was strong, which is in line with the previous findings that Ang-2 is produced and expressed mainly by the endothelial cells. Ang-2 has been considered to function as a Tie-2 antagonist thus promoting tumor angiogenesis and inflammation by competing with Ang-1 for Tie-2 binding and inhibiting Tie-2 activity. The blockade of Tie-2 signaling is thought to lead to the loss of pericyte wall, vessel destabilization, which induces VEGF-dependent angiogenesis. However, Daly et al. found that Ang-2 blockade inhibited tumor growth by decreasing Tie-2 activity indicating that Ang-2 could function as a Tie-2 agonist in tumor xenograft models [27, 28].

According to our previously published data [17], the level of Ang-2 PP correlated significantly to that of VEGF-A PP (r = 0.3, p = 0.014) in high grade tumors. This supports the concept that tumor derived VEGF-A interacts with the Ang/ Tie-2 system by acting in a paracrine manner to induce the remodeling (vessel regression) of the host vasculature and stimulating neoangiogenesis to support tumor growth and invasion [12, 27]. Unexpectedly the level of Ang-2 immunoexpression was inversely correlated to the mRNA levels (S2 Table in S1 File). This can be explained by the distribution of Ang-2 in tumor samples: IRS and PP represent the cytoplasmic expression of the tumor epithelial cells, not stromal or endothelial expression, whereas the mRNA specimen does not distinguish between the different cellular locations. It can also be influenced by the post-translational regulation of the protein synthesis.

In our study, Tie-2 was mainly weakly expressed in epithelial tumor cells of primary cancer. Strong membranous staining was evident in some cases and vascular endothelium and stromal fibroblasts were positive in most cases, as has been observed in two publications investigating ovarian cancer [12, 15]. Tie-2 has been shown to be overexpressed in gastric, breast and hepatocellular cancer tissues where it was related to tumor angiogenesis and to a worse clinical outcome in gastric and breast cancers [29]. Tie-2 is primarily expressed by the endothelial cells, but it has also been identified in non-stromal cancer cells of leukemia, thyroid and breast cancers. In gastric cancer, it was mainly detected in the cytoplasm of stromal macrophages and in gastric cancer cells. Yang et al. stated that Tie-2 expression was increased in the hypoxic regions of the gastric cancer and the state of tumor hypoxia in solid tumors was a marker for a poor prognosis. In our study, strong expression of Tie-2 in primary high grade serous and endometrioid ovarian cancer predicted a significantly shorter overall survival in both univariate and multivariate survival analyses. In addition, in breast cancer, the tumor immunoexpression of Tie-2 has been linked to a poor OS in a univariate analysis [30].

We showed that Tie-2 overexpression was a prognostic factor for poor survival. There is also evidence that a combined Tie-2 targeted gene therapy exerted an antitumor effect in animal models. Co-targeting of VEGFR1, -3 and Tie-2 has been associated with reduced growth of solid human ovarian cancer in mice, and furthermore, combined gene therapy using VEGFR2 and Tie-2 with chemotherapy has reduced the growth of tumors, as well as diminishing the formation of ascites in mice [31].

Expression of Ang-2 and Tie-2 was significantly stronger in related omental metastases than in primary high grade serous ovarian cancers indicating that in ovarian cancer, they play

a role in the metastatic process. In tumor angiogenesis models, it has been suggested that angiogenesis is more active at the tumor periphery, enabling the invasion and dissemination whereas in the centre of aggressively growing solid tumors, the balance is dominated by vessel regression and the development of necrosis [27]. Additionally, in mice, Ang-2, as well as Ang-4, has been shown to promote invasion of ovarian cancer cells into host peritoneal and liver tissue parenchyma [15]. This could also explain the more active angiogenesis in distal metastasis. In our earlier study, we detected higher expressions of proangiogenic VEGF-A, VEGF-D and VEGFR1 in distal metastases as compared to primary high grade serous ovarian tumors, which may be evidence for the complementary role of the Ang-2/ Tie-2 system [17]. In fact, this finding supports the current concept of ovarian cancer treatment, i.e. using antiangiogenic therapy only in disseminated cancer or after incomplete primary surgical resection [7].

However metastasis is a complex process involving several factors including the degradation of the extracellular matrix, the epithelial-to-mesenchymal transition, tumor angiogenesis and the development of inflammatory tumor microenvironment and defects in programmed cell death [32]. In fact, omental metastasis occurs by exfoliation of malignant cells into the peritoneal cavity, ascites formation and implantation of cancer cells especially to the omental surface [33]. According to the fact, that tumors can grow without vascular supply only to the size of $1–2 \text{ mm}^3$, omental metastasis is not an exception and therefore also need angiogenic factors to survive [34]. In breast and lung cancer, it has been shown that the role of Ang-2 is not only stimulating angiogenesis, but promoting invasion and providing mechanism for tumor cells to acquire an invasive phenotype by inducing epithelial to mesenchymal transition (EMT) and thus contributing metastasis. This could also explain higher Ang-2 and Tie-2 in metastasis [24, 35]. This dualistic contribution of Ang-2 in metastasis can explain its higher expression in omental tumors. To more closely evaluate the role of angiopoietins and Tie-2 in accordance with EMT markers during metastatic process needs further studies.

The Tie-1 receptor was expressed very weakly in tumor epithelial cells of ovarian cancer patients. In addition, lower expression was associated with larger residual tumour after primary surgery and the recurrence of the cancer in serous tumors as described above for Ang-2. Expression of Tie-1 was detected mainly in stromal fibroblasts and vascular endothelium. The function of Tie-1 is less well understood mainly because of its orphan receptor status. It has been claimed to have context-dependent effects on Tie-2 expression in tumor angiogenesis, counter-regulate Tie-2 in angiogenic tip cells and sustain Tie-2 signaling in remodeling stalk cell vasculature [36]. Although we did not find any correlation to patient survival, Tie-1 expression was reported to exhibit a negative correlation with 5-year survival in gastric cancer patients [37].

Van Cutsem et al. have demonstrated that combination therapy with Ang-2 specific antibody and VEGF blocker aflibercept more strongly reduced tumour growth than either agent on its own [38]. Anti-angiopoietin therapy with trebananib binds both Ang-1 and Ang-2, preventing the interaction with Tie-2. In a phase 3 randomised trial in patients with recurrent ovarian cancer, median PFS was significantly longer in the trebananib group as compared to the placebo [9, 39, 40]. However, in the first line study of ovarian cancer published recently, trebananib did not improve PFS [41]. This may partly be explained by our result, i.e. Ang-2 was not a prognostic factor for poor survival.

There is an urgent need to devise predictive biomarkers to optimize the use of antiangiogenic therapies; for example, in ovarian cancer the activity of bevacizumab has only been observed in high risk cases [42, 43]. The elevated plasma concentrations of Tie-2 have been shown to predict the progression of ovarian and colorectal cancer after treatment with the anti-VEGF, bevacizumab [44, 45].

It is essential to clarify the ongoing pathological processes at the tissue level during the progression of ovarian cancer in order to develop reliable prognostic biomarkers and optimal

therapy. This study shows that the expression of Tie-2 receptor in ovarian cancer was associ-ated with a poor prognosis in a multivariate OS analysis. In addition, Tie-2 and Ang-2 were more strongly expressed in related distal metastatic tumors, potentially reflecting the more active angiogenesis in metastases. Furthermore, low expression levels of Ang-2 and Tie-1 in ovarian cancer cells were linked to an aggressive phenotype of ovarian cancer cells.

To conclude, the results of this study support the use of Tie-2 receptor as a biomarker in high grade serous ovarian cancer as its expression is significantly linked to OS and its expres-sion is higher in metastatic lesions.

## Supporting information

**S1 Fig.** Positive and negative tissue control stainings for a) Ang-2 positive in ovarian tissue, c) Tie-1 positive in tonsillar tissue, e) Tie-2 positive in ovarian carcinoma, b), d) and f) are corre-sponding Ang-2, Tie-1 and Tie-2 ovarian carcinoma negative control stainings (primary anti-body omitted).
(PDF)

**S1 Dataset.**
(XLSX)

**S1 File.**
(DOCX)

## Acknowledgments

We thank Mrs Helena Kemiläinen for skillful technical assistance and Mr Tuomas Selander for assistance in statistics.

## Author Contributions

**Conceptualization:** Minna Sopo, Hanna Sallinen, Maarit Anttila.

**Data curation:** Hanna Sallinen, Maarit Anttila.

**Funding acquisition:** Minna Sopo, Hanna Sallinen.

**Investigation:** Minna Sopo, Annukka Kivelä.

**Methodology:** Maarit Anttila.

**Project administration:** Seppo Ylä-Herttuala.

**Resources:** Seppo Ylä-Herttuala, Veli-Matti Kosma.

**Supervision:** Hanna Sallinen, Kirsi Hämäläinen, Leea Keski-Nisula, Maarit Anttila.

**Validation:** Kirsi Hämäläinen.

**Visualization:** Minna Sopo.

**Writing – original draft:** Minna Sopo.

**Writing – review & editing:** Hanna Sallinen, Leea Keski-Nisula, Maarit Anttila.

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
