## [Decision Letter · Decision Letter 0]

6 Jan 2020

PONE-D-19-33270

HIGH EXPRESSION OF TIE-2 PREDICTS POOR PROGNOSIS IN PRIMARY HIGH GRADE SEROUS OVARIAN CANCER

PLOS ONE

Dear Dr. Sopo,

Thank you for submitting your manuscript to PLOS ONE. After careful consideration, we feel that it has merit but does not fully meet PLOS ONE’s publication criteria as it currently stands. Therefore, we invite you to submit a revised version of the manuscript that addresses the points raised during the review process.

Specifically, the authors should include controls, improve quality of figures and consistency of the presentation of IHC data, details of IHC and editorial corrections including sentences.  

We would appreciate receiving your revised manuscript by Feb 20 2020 11:59PM. To enhance the reproducibility of your results, we recommend that if applicable you deposit your laboratory protocols in protocols.io, where a protocol can be assigned its own identifier (DOI) such that it can be cited independently in the future. For instructions see: http://journals.plos.org/plosone/s/submission-guidelines#loc-laboratory-protocols

We look forward to receiving your revised manuscript.

Kind regards,

Surinder K. Batra

Academic Editor

PLOS ONE

Journal Requirements:

2.  Please provide additional details regarding participant consent. In the ethics statement in the Methods and online submission information, please ensure that you have specified (1) whether consent was informed and (2) what type you obtained (for instance, written or verbal, and if verbal, how it was documented and witnessed). If your study included minors, state whether you obtained consent from parents or guardians. If the need for consent was waived, please ensure that you have discussed whether all data were fully anonymized before you accessed them and/or whether the IRB or ethics committee waived the requirement for informed consent”.

3. In your Methods section, please provide additional information about the participant recruitment method and the demographic details of your participants. Please ensure you have provided sufficient details to replicate the analyses such as: a) the recruitment date range (month and year), b) a table of relevant demographic details and c) a description of how participants were recruited.

4. We suggest you thoroughly copyedit your manuscript for language usage, spelling, and grammar. If you do not know anyone who can help you do this, you may wish to consider employing a professional scientific editing service.  

Additional Editor Comments (if provided):

Reviewers' comments:

Reviewer's Responses to Questions

**Comments to the Author**

1. Is the manuscript technically sound, and do the data support the conclusions?

Reviewer #1: No

Reviewer #2: No

2. Has the statistical analysis been performed appropriately and rigorously? 

Reviewer #1: Yes

Reviewer #2: No

3. Have the authors made all data underlying the findings in their manuscript fully available?

Reviewer #1: Yes

Reviewer #2: Yes

4. Is the manuscript presented in an intelligible fashion and written in standard English?

Reviewer #1: Yes

Reviewer #2: No

5. Review Comments to the Author

Reviewer #1: In this manuscript, the authors investigate the expression of Ang-2, Tie-1 and Tie 2 in ovarian cancer and their relationship with patient survival, chemo resistance and other clinical parameters. The main conclusion of the study is that low expression of Tie-2 is associated with increased survival of ovarian cancer patients. High expression of Tie-2 was also detected in metastasized ovarian tumors. Surprisingly, Ang-2 expression was low in ovarian tumors and predicted poor overall survival. There is limited data in the literature on the expression of these angiogenic markers in ovarian cancer. The study therefore adds to our understanding of ovarian cancer biology and may also support the targeting of Tie-2 and Ang-1. There are several shortcomings of this manuscript that reduce enthusiasm.

1. Immunohistology is the primary method used to monitor the expression of the angiogenesis markers in ovarian tissues. Therefore, it is important that the IHC data be of high quality. The expression of Tie-2 is diffuse and in some of the images provided by the authors, it is not clear if the expression is sufficiently specific. Authors are requested to demonstrate antibody specificity. At the same time, this study will also benefit from the inclusion of positive controls of human tissues that are known to be positive for Tie-2, Ang-2 and the other markers used in this study.

2. Authors have used the immunoreactivity score (IRS) which takes into account the percent of positive cells and eth intensity of the staining. It is therefore confusing as to why the authors selectively show IRS data in some figures and positive percentage (PP) and specific intensity (SI) for others. It will be best if authors use IRS data for the semi-quantitative analysis. Alternatively, authors may also conduct the quantitation using the accepted H-scoring system.

3. There is no correlation between IHC expression and mRNA expression. Authors point out that this is likely because the RNA samples are derived from the entire tissue and not from cancer, endothelial or stromal cells. This logic makes sense. But it also implies that there is no necessity to include the qPCR results as they only confuse the results.

4. Authors should specify if the samples used for IHC were obtained from treatment naïve patients.

5. Overall, the results are presented in a confusing manner. Often times, in the results section, the number of the figure or the table is not explicitly mentioned. Therefore, it is hard for the reader to determine which data the authors are referring to in the text.

Reviewer #2: This manuscript is focused on analyzing the impact of TIE-2 on the prognosis of serous ovarian cancer. To prove the concept, the authors used 86 primary epithelial ovarian cancer samples and 16 omental metastasis. They observed high epithelial expression of Tie-2 in primary high-grade serous ovarian cancer, which is correlating with survival. Angiopoietin-2 showed low expression in serous ovarian tumors and associated with shorter survival. Additionally, the expression of angiopoietin-2 and Tie-2 showed increased expression in distal omental metastasis compared to primary tumors.

This manuscript needs a major revision for publication.

• Several sentences were incomplete in the manuscript, so extensive language correction is required.

• There is no clear result given the angiopoietin-2 and Tie-2 in other subtypes shown in sample numbers.

• The expression of angiopoietin-2 and Tie-2 in distal omental metastasis is showing an interesting result; however, the discussion is missing.

• Since it is correlated with the omental metastasis, what is the status in other metastasis?

• There is no statistical calculation shown in qRT-PCR of all the molecules. How many mRNA samples were used for this analysis, and how many times, experiments were repeated to calculate significance? Because this is the base result for the manuscript.

• There is no H-score calculated for the intensity of Tie-1, two, and Ang 2 staining in serous ovarian cancer samples. What is the subcellular localization of these molecules?

• Immunohistochemistry images are showing poor quality of staining.

6. PLOS authors have the option to publish the peer review history of their article (what does this mean?). If published, this will include your full peer review and any attached files.

Reviewer #1: No

Reviewer #2: No

---

## [Author Response · Author response to Decision Letter 0]

19 Feb 2020

Response to Academic Editor

We thank Academic Editor for the valuable comments. Each of your comment or question is indicated in bold followed by our response. All changes are also highlighted in the “Revised manuscript with track changes” and corrected in the unmarked version of revised manuscript without track changes.

1. Specifically, the authors should include controls, improve quality of figures and consistency of the presentation of IHC data, details of IHC and editorial corrections including sentences. 

We included 30 IHC control tissue samples in this study: 22 positive and 8 negative controls of human tissues (adnexal, uterine and tonsillar samples) positive for studied markers. 

(added in Materials and Methods, Immunohistochemistry, page 8, lines 151-55)

We improved the quality of figures. Figure 2 (Figure 3 in the first version of the manuscript) is made newly in order to clear the image. Former Figure 2 was removed. Also Figures 1 and 3 were made newly to improve the consistency of the presentation and clarify the IHC data. 

We removed the positive percentage (PP) and staining intensity (SI) parameters from the manuscript, tables (2 and 3, page 14-16, lines 299-307) and the figures (1 and 3) and used only immunoreactive score (IRS) data for the semi-quantitative analysis to improve the presentation of the details of IHC data. We left the parameters in the supplementary table 2 and 3 to clarify which parameters explain the result. 

The English language and editorial corrections for this manuscript was made by Dr Ewen MacDonald. 

2. To enhance the reproducibility of your results, we recommend that if applicable you deposit your laboratory protocols in protocols.io, where a protocol can be assigned its own identifier (DOI) such that it can be cited independently in the future.

Our laboratory protocols are the standard, official, qualified protocols used in the Department of Pathology at the University of Eastern Finland. The protocols are openly delivered, but not available in English at this point. So we will work on that and will deposit the protocols in protocols.iu in future. 

We have ensured that the manuscript meets PLOS ONE`s style requirements, including file naming.

4. Please provide additional details regarding participant consent. In the ethics statement in the Methods and online submission information, please ensure that you have specified (1) whether consent was informed and (2) what type you obtained (for instance, written or verbal, and if verbal, how it was documented and witnessed). If your study included minors, state whether you obtained consent from parents or guardians. If the need for consent was waived, please ensure that you have discussed whether all data were fully anonymized before you accessed them and/or whether the IRB or ethics committee waived the requirement for informed consent”.

The written, informed consent was obtained from the patients participating in this study. Methods page 8, lines 143-44.

5. In your Methods section, please provide additional information about the participant recruitment method and the demographic details of your participants. Please ensure you have provided sufficient details to replicate the analyses such as: a) the recruitment date range (month and year), b) a table of relevant demographic details and c) a description of how participants were recruited. 

All patients were Finnish, white females (Caucasian), median age 58 years (26-88). They were recruited in the gynecological policlinic, when they were sent for the diagnostics and treatment of ovarian tumors. Recruitment date range was 1.9.1999 �28.3.2007. 

Methods page 5, lines 113, 123-24.

6. We suggest you thoroughly copyedit your manuscript for language usage, spelling, and grammar. If you do not know anyone who can help you do this, you may wish to consider employing a professional scientific editing service.

The English language corrections and copyediting for this manuscript was made by Dr Ewen MacDonald. 

A point-by-point Response letter to the Reviewers

Response to Reviewer 1

We thank Reviewer for the valuable comments. Each of your comment or question is indicated in bold followed by our response. All changes are also highlighted in the “Revised manuscript with track changes” and corrected in the unmarked version of revised manuscript without track changes.

1. Immunohistology is the primary method used to monitor the expression of the angiogenesis markers in ovarian tissues. Therefore, it is important that the IHC data be of high quality. The expression of Tie-2 is diffuse and in some of the images provided by the authors, it is not clear if the expression is sufficiently specific. Authors are requested to demonstrate antibody specificity. At the same time, this study will also benefit from the inclusion of positive controls of human tissues that are known to be positive for Tie-2, Ang-2 and the other markers used in this study.

Immunohistochemical stainings have been implemented according to the official protocols by the Department of Pathology at the University of Eastern Finland by using the antibodies of Ang-2 by Novus, Tie-1 by Abcam and Tie-2 by Santa Cruz as described in the Materials and Methods Immunohistochemistry section (page 8, lines 147-51).

The antibodies used have been checked by the vendors and the Western blots have been made. In addition to confirm the expression of growth factors and receptors we performed qRT-PCR to see their expression in the RNA-level. 

We included 30 IHC control tissue samples in this study: 22 positive and 8 negative controls of human tissues (adnexal, uterine and tonsillar samples) positive for aforementioned markers. 

(added in Materials and Methods, Immunohistochemistry, page 8, lines 151-55)

Figure 2 (Figure 3 in the first version of the manuscript) is made newly in order to clear the image. Former Figure 2 was removed.

2. Authors have used the immunoreactivity score (IRS) which takes into account the percent of positive cells and the intensity of the staining. It is therefore confusing as to why the authors selectively show IRS data in some figures and positive percentage (PP) and specific intensity (SI) for others. It will be best if authors use IRS data for the semi-quantitative analysis. Alternatively, authors may also conduct the quantitation using the accepted H-scoring system.

We removed the positive percentage (PP) and staining intensity (SI) parameters from the manuscript, tables (2 and 3, page 14-16, lines 299-307) and the figures (1 and 3) and used only immunoreactive score (IRS) data for the semi-quantitative analysis. We left the parameters in the supplementary table 2 and 3 to clarify which parameters explain the result. 

We did not use H-score, because it is used to evaluate the nuclear positivity.

3. There is no correlation between IHC expression and mRNA expression. Authors point out that this is likely because the RNA samples are derived from the entire tissue and not from cancer, endothelial or stromal cells. This logic makes sense. But it also implies that there is no necessity to include the qPCR results as they only confuse the results.

We removed the qRT PCR results from the manuscript. We would like to leave it in the supplementary material (S4 Table), because it is considered the other base result of the study.

4. Authors should specify if the samples used for IHC were obtained from treatment naïve patients.

Added in Materials and Methods, Patients and data collection, page 5, line 114-15.

5. Overall, the results are presented in a confusing manner. Often times, in the results section, the number of the figure or the table is not explicitly mentioned. Therefore, it is hard for the reader to determine which data the authors are referring to in the text.

We defined the presentation of the results, specified and added the references to the figures and tables.

Response to Reviewer 2

We thank Reviewer for the valuable comments. Each of your comment or question is indicated in bold followed by our response. All changes are also highlighted in the “Revised manuscript with track changes” and corrected in the unmarked version of revised manuscript without track changes.

1. Several sentences were incomplete in the manuscript, so extensive language correction is required.

English language correction has been made for this manuscript in January 2020 by Dr Ewen MacDonald.

2. There is no clear result given the angiopoietin-2 and Tie-2 in other subtypes shown in sample numbers.

The number of other subtypes than high grade serous and endometrioid carcinoma is very small (mucinous tumors 11 and clear cell 5, others 4), so making a definitive conclusions or giving exact statistical results of small subgroups is not relevant. 

3. The expression of angiopoietin-2 and Tie-2 in distal omental metastasis is showing an interesting result; however, the discussion is missing.

We have discussed about the significance of the higher expression of the metastases in Discussion page 18-19, lines 373-85. 

4. Since it is correlated with the omental metastasis, what is the status in other metastasis?

We included only omental metastases in this study, since this type of dissemination is typical for the high grade ovarian cancer. To evaluate the other metastases, could be the idea for future studies. 

5. There is no statistical calculation shown in qRT-PCR of all the molecules. How many mRNA samples were used for this analysis, and how many times, experiments were repeated to calculate significance? Because this is the base result for the manuscript.

Samples from 22 primary and 15 metastatic tumors were analysed by qRT-PCR for each antibody (total 66 primary and 45 metastatic samples). Each sample has been repeated three times to calculate the significance. (Materials and Methods, qRT-PCR, page 9, line 174)

Only the significant correlations between the immunohistochemistry and the mRNA levels are shown in the Supplementary table 4. 

6. There is no H-score calculated for the intensity of Tie-1, two, and Ang 2 staining in serous ovarian cancer samples. What is the subcellular localization of these molecules?

We did not use H-score, because it is used to evaluate the nuclear positivity. Ang-2 expression was detected mainly in the cytoplasm of the tumor epithelial cells and vascular endothelial cells. Tie-2 stained most in the cytoplasm and the membranes of the tumor epithelial cells, vascular endothelium and particularly in metastases stromal fibroblasts. Tie-1 expression was practically not seen in the tumor epithelial cells, it was strongest in the tumor stromal cells. (Results pages 10-11, lines 202-08, 222-33)

7. Immunohistochemistry images are showing poor quality of staining.

Figure 2 (Figure 3 in the first version of the manuscript) is made newly in order to clear the image. Former Figure 2 was removed.

---

## [Decision Letter · Decision Letter 1]

17 Apr 2020

PONE-D-19-33270R1

HIGH EXPRESSION OF TIE-2 PREDICTS POOR PROGNOSIS IN PRIMARY HIGH GRADE SEROUS OVARIAN CANCER

PLOS ONE

Dear Dr Sopo,

Thank you for submitting your manuscript to PLOS ONE. After careful consideration, we feel that it has merit but does not fully meet PLOS ONE’s publication criteria as it currently stands. Therefore, we invite you to submit a revised version of the manuscript that addresses the points raised during the review process.

As only one of the original reviewers was available, two additional reviewers have also read the manuscript. Please address the comments of all reviewers. We would appreciate receiving your revised manuscript by Jun 01 2020 11:59PM. To enhance the reproducibility of your results, we recommend that if applicable you deposit your laboratory protocols in protocols.io, where a protocol can be assigned its own identifier (DOI) such that it can be cited independently in the future. For instructions see: http://journals.plos.org/plosone/s/submission-guidelines#loc-laboratory-protocols

We look forward to receiving your revised manuscript.

Kind regards,

Elizabeth Christie

Academic Editor

PLOS ONE

Reviewers' comments:

Reviewer's Responses to Questions

**Comments to the Author**

1. If the authors have adequately addressed your comments raised in a previous round of review and you feel that this manuscript is now acceptable for publication, you may indicate that here to bypass the “Comments to the Author” section, enter your conflict of interest statement in the “Confidential to Editor” section, and submit your "Accept" recommendation.

Reviewer #1: All comments have been addressed

Reviewer #3: (No Response)

Reviewer #4: (No Response)

2. Is the manuscript technically sound, and do the data support the conclusions?

Reviewer #1: Yes

Reviewer #3: Partly

Reviewer #4: No

3. Has the statistical analysis been performed appropriately and rigorously? 

Reviewer #1: Yes

Reviewer #3: Yes

Reviewer #4: No

4. Have the authors made all data underlying the findings in their manuscript fully available?

Reviewer #1: Yes

Reviewer #3: No

Reviewer #4: No

5. Is the manuscript presented in an intelligible fashion and written in standard English?

Reviewer #1: Yes

Reviewer #3: Yes

Reviewer #4: No

6. Review Comments to the Author

Reviewer #1: Authors have addressed previous critique.

The figures are of good quality and the manuscript has been suitably modified to highlight relevance accurately.

Reviewer #3: 1. The manuscript thematically purports to investigate biomarkers of tumour angiogenesis in ovarian cancer. Ang-2 and Tie-2 protein levels assessed by immunohistochemistry are shown to relate to overall survival; however the predominant expression of both of these proteins is in epithelial tumour cells. Although the predictive value of these markers appears to be strong, it is unclear how this relates to tumour angiogenesis - microvessel density is not assessed in this study. Furthermore, while the Discussion maintains the focus on Ang2/Tie2 in tumour angiogenesis and metastasis, it appears to ignore studies in other tumour types showing that Ang2 signaling through integrins on tumour cells can cell-intrinsically promote invasion and metastasis by inducing epithelial to mesenchymal transition (EMT; e.g. Imanishi et al. 2007 Cancer Research 67(9):4254–63; Dong et al. 2018 Oncotarget 9:12705-12717, and others). This alone could explain the observation of higher Ang2 (and possibly Tie2) expression in metastases than in primary tumours. The authors should acknowledge promotion of EMT by Ang2 as a possible mechanism, and acknowledge the limitations of their data in making conclusions regarding tumour angiogenesis and systemic metastasis.

2. In Figure 1 it is not clear what values are displayed – are these median values of IRS? Epithelial, stromal or overall? Presuming that an IRS value was calculated for each tumour specimen, and the data are non-normally distributed, the data could be better presented as a box-and-whisker plot (or similar) with individual data points. Currently there is no indication of the spread of the data.

3. Please provide images of the positive and negative tissue controls for Ang-2, Tie-2 and Tie-1 as Supporting Information, along with references indicating expected patterns of staining in positive control tissues for each protein. It is not sufficient merely to state that the negative controls were negative. Were antibody isotype controls included? If Western Blots have been performed to validate the specificity of the antibodies used for IHC, please present this data as well.

4. Some of the data concerning Ang-2 expression are counterintuitive. Low Ang-2 expression is associated with poor overall survival in spite of higher Ang-2 immunoreactivity in metastases, and in contrast with studies in other tumour types showing that Ang-2 promotes tumour invasiveness (ref. 22). The negative correlation between Ang-2 immunoreactivity and mRNA levels is also perplexing. Although there are potential explanations for these discrepancies, it does raise suspicion about the specificity of the Ang-2 antibody used, and underscores the importance of independent validation of antibody specificity by investigators. Furthermore, the authors state in the Abstract that relative expression of Ang-2 determined by qRTPCR correlated significantly with IHC protein levels, which is misleading – the correlation is statistically significant, but negative.

5. Line 235: the authors state that “the correlation [between Ang-2 and Tie-1] was significant among other histological types”. If so, please present r and p values in the text or a table. Please clarify in the Methods section which statistical test was used to measure this correlation.

6. In Tables 2 and 3, the authors seem to only present data for analyses that yielded statistically significant results. Did all analyses performed yield statistically significant results? Please provide the results of all analyses for completeness. Were all parameters analysed for their relationship to overall survival also assessed regarding PFS? If so, please present these data as well.

7. Similarly, if correlation analyses between IRS and qRTPCR results were performed for Tie1 and Tie2, please present these as well. Given that the authors have removed most of the qRTPCR data from the previous version of their manuscript, they should consider toning down claims of their study’s importance/uniqueness related to this.

8. The authors state (line 97) that “there are no studies investigating the tissue expressions of Ang-2 and its receptors in ovarian cancer”, referencing Table S1, yet this Table contains several examples where such investigations appear to have been published, e.g. ref. 19. Please revise this statement.

9. Table S1: several references within this table do not appear within the main References list. Please add a list of full Supplementary references to the Supporting information file to cover these.

10. Statistical analysis: I am not familiar with the use of a Mann-Whitney U test as a post-hoc test for pairwise comparisons subsequent to the overarching Kruskal-Wallis test. In my understanding Dunn’s post-hoc test is more appropriate. Please provide a reference validating the appropriateness of this approach, or seek expert statistical advice.

11. Do the omental metastases examined reflect systemic or local/in transit metastasis?

12. S1 Table 1st row data, 3rd column: “Post-oper” – word incomplete

13. Line 326: “angiogenetic” should be “angiogenic”

Reviewer #4: 1) Figure 1 – Figure 1 and the Figure 1 legend both lack sufficient detail to interpret the immunohistochemistry data. Readers should be able to determine the distribution of staining scores across the cohort, which is not possible with data presented as a histogram. Please change histograms to violin plots, which will allow you to show the median score in each group, the individual values per patient, their distribution and the density. The figure or legend must also state the number of patients included in each group. A statistical test should also be used to compare expression values for each marker between primary tumors and metastases, and P values should be reported for each comparison. The y-axis also should be labelled. Please also state in the legend whether the IRS was calculated from the tumor epithelium or stroma.

2) Results – Throughout the text the authors refer to staining patterns in primary high grade serous and endometrioid cancer. This data is not shown. For example, lines 205-206 – The authors report that “the stromal staining of Ang-2 was mostly weak in 74% of cases in primary high grade serous and endometrioid tumors”. No Figure or Table is referred to. When referring to staining intensity, perhaps the authors mean to refer to Table S2? However, Table S2 contains staining intensity for all histotypes combined. Where is the data presented for selected primary high grade serous and endometrioid tumors?

3) Results, lines 224-225 – “stromal staining was strong in 64% of cases, emphasizing stromal fibroblasts and vascular endothelium (Figs 1 and 2c-d)”. Figures 1 and 2 do not show that 64% of cases have strong stromal staining. Authors need to carefully check that each results statement is correctly linked to the appropriate Figure and/or Table, and that the Figures and/or Tables referred to actually contain that data.

4) Table S2 – Given the amount of key data presented in Table S2, this should be changed to a main Table within the manuscript.

5) Table S2 – It is not clear whether the P values are calculated for just the paired primary and metastatic samples from the same patients, or unpaired groups of all primary and all metastases. Consider doing both. The range of IRS values are missing from the table.

6) Results, lines 234-237 – “The level of Tie-1 expression was moderately, but significantly, correlated with its ligand Ang-2 (r = 0.5, p < 0.001) in primary serous ovarian tumors, and the correlation was also significant among other histological types. The expression level of Tie-2 showed a weak, but significant correlation with Ang-2 (r = 0.3, p = 0.012) and moderate correlation with Tie-1 (r = 0.5, p < 0.001) in primary cancer.” Which values were used to test these correlations and in which cellular compartment?

7) Results, lines 246-247 – “The level of Tie-1 expression did not differ between primary and metastatic lesions (Fig 1, 2 and S2 Table)”. However table S2 shows that the stromal expression of Tie-2 does differ between primary and metastatic lesions (p-value 0.046. Please clarify.

8) Table S3 – Please indicate whether the values were calculated from epithelial or stromal cells.

9) Results, lines 251-270 – Throughout this section the authors are investigating the association between biomarker expression and various clinical parameters. However, we are not shown the expression data. For example, “The low level epithelial expression of Ang-2 in primary tumors associated significantly with the ovarian cancer recurrence and with the greater residual tumor size (� 1cm) after primary surgery (p = 0.018, p = 0.012)”. For this comparison, the authors should show the median IRS for tumours with residual tumor > 1cm, compared to IRS of tumours with residual tumor < 1cm and no residual tumour. Therefore, in addition to the P values in Table S3, please include the median IRS and PP values for each group comparison.

10) Results, lines 254-256 – “In high grade serous tumors, high levels of Ang-2 staining were related to the resistance to platinum chemotherapy (p = 0.017) (S3 Table).” Platinum resistance and high-grade serous tumors are both not shown in Table S3.

11) Results, line 284-286 – The authors need to explain how high Tie-2 is associated with shorter overall survival, and low Ang-2 is associated with poor survival, and yet Tie-2 and Ang-2 expression are positively correlated (as stated in Results lines 236-237). Presumably these are different populations of patients? Please explore and explain this result.

12) Figure 3 – In the figure, please indicate the number of individuals at risk at each major time interval under each Kaplan-Meier plot.

13) Figure 3 and Table 3 – How was high and low assigned for each biomarker, and what was the cut-off used for high and low?

14) qRT-PCR is mentioned in the Abstract, the Methods and Discussion, however not mentioned in the Results. Please clarify.

15) Table S4 – Please include a table description indicating that Ang-2 IRS and PP is being correlated with qRT-PCR, and how the P values were calculated, and the numbers of samples tested. Also, please include the data for Tie-1 and Tie-2.

16) Grammar needs to be corrected throughout the manuscript.

17) Table 3 – Table 3 shows 48 patients with low Ang-2 IRS and 82 patients with high Ang-2, which added together equals 130 patients. However, there are only 86 women described in the Materials and Methods. Please clarify. Also, Tie-1 is missing from this Table.

7. PLOS authors have the option to publish the peer review history of their article (what does this mean?). If published, this will include your full peer review and any attached files.

Reviewer #1: No

Reviewer #3: No

Reviewer #4: No

---

## [Author Response · Author response to Decision Letter 1]

29 May 2020

A point-by-point Response letter to the Reviewers

Response to Reviewer 3

We thank Reviewer for the valuable comments. Each of your comment or question is indicated in bold followed by our response. All changes are also highlighted in the “Revised manuscript with track changes” and corrected in the unmarked version of revised manuscript without track changes.

1. The manuscript thematically purports to investigate biomarkers of tumour angiogenesis in ovarian cancer. Ang-2 and Tie-2 protein levels assessed by immunohistochemistry are shown to relate to overall survival; however the predominant expression of both of these proteins is in epithelial tumour cells. Although the predictive value of these markers appears to be strong, it is unclear how this relates to tumour angiogenesis - microvessel density is not assessed in this study. Furthermore, while the Discussion maintains the focus on Ang2/Tie2 in tumour angiogenesis and metastasis, it appears to ignore studies in other tumour types showing that Ang2 signaling through integrins on tumour cells can cell-intrinsically promote invasion and metastasis by inducing epithelial to mesenchymal transition (EMT; e.g. Imanishi et al. 2007 Cancer Research 67(9):4254–63; Dong et al. 2018 Oncotarget 9:12705-12717, and others). This alone could explain the observation of higher Ang2 (and possibly Tie2) expression in metastases than in primary tumours. The authors should acknowledge promotion of EMT by Ang2 as a possible mechanism, and acknowledge the limitations of their data in making conclusions regarding tumour angiogenesis and systemic metastasis.

Added the acknowledgement of promotion of EMT by Ang-2 as a possible mechanism of its higher expression in metastasis compared to the primary tumors and the limitation of this data.

Discussion, page 21, rows 401-406.

Added references 22 and 31. References, page 25, rows 525-27 and 26, rows 554-56 .

2. In Figure 1 it is not clear what values are displayed – are these median values of IRS? Epithelial, stromal or overall? Presuming that an IRS value was calculated for each tumour specimen, and the data are non-normally distributed, the data could be better presented as a box-and-whisker plot (or similar) with individual data points. Currently there is no indication of the spread of the data.

Figure 1 is now presented as a violin blot with individual data points, describing both the median values of IRS and the density of the distribution of the data points in each group.

IRS is calculated from the staining intensity and the percentage of positively stained cells of the tumor epithelial cells. Methods page 8, lines 159-66.

3. Please provide images of the positive and negative tissue controls for Ang-2, Tie-2 and Tie-1 as Supporting Information, along with references indicating expected patterns of staining in positive control tissues for each protein. It is not sufficient merely to state that the negative controls were negative. Were antibody isotype controls included? If Western Blots have been performed to validate the specificity of the antibodies used for IHC, please present this data as well.

Images of the positive and negative tissue controls for Ang-2, Tie-1 and Tie-2 are now included as Supporting Information (S1 Figure). References indicating expected patterns of staining in positive control tissues are also provided for each protein.

Methods page 8, lines 148-53; References 14-16, page 25. 

Western blots were not included in this study. We believe that with immunohistochemical detection, we were able to detect the expression of growth factor and its receptors reliably, and analyse the exact location of these proteins in tumor tissue. The vendors of the antibodies have already checked and performed the Western blots for each antibody. In addition, to confirm the expression of growth factors and their receptors we performed qRT-PCR to see their expression in RNA level. 

4. Some of the data concerning Ang-2 expression are counterintuitive. Low Ang-2 expression is associated with poor overall survival in spite of higher Ang-2 immunoreactivity in metastases, and in contrast with studies in other tumour types showing that Ang-2 promotes tumour invasiveness (ref. 22). The negative correlation between Ang-2 immunoreactivity and mRNA levels is also perplexing. Although there are potential explanations for these discrepancies, it does raise suspicion about the specificity of the Ang-2 antibody used, and underscores the importance of independent validation of antibody specificity by investigators. Furthermore, the authors state in the Abstract that relative expression of Ang-2 determined by qRTPCR correlated significantly with IHC protein levels, which is misleading – the correlation is statistically significant, but negative.

There are different results of the significance of Ang-2 expression on survival in different cancer types. Zhang L (ref.19) thought that Ang-2 is detectable in tumor cells in only 12% of tumor specimens of ovarian cancer. In contrast Brunckhorst (ref. 20) found that tumor cells commonly expressed Ang-2. In these earlier studies the correlation to the clinical factors has not been found in contrast to this study. In this study the low Ang-2 expression associated with poor overall survival is specifically the tumor epithelial cell expression in primary tumors! The metastases were not taken into count in this analysis. 

In addition, low Ang-2 expression in primary tumors was associated with poor survival in univariate analysis, not in multivariate analysis. Other tissues have also been stained with these same antibodies with positive expression results in the laboratory of the Institute of Pathology in University of Eastern Finland.

The negative correlation of Ang-2 to the mRNA levels could be explained by the influence of the post-translational regulation of the protein synthesis and the distribution of Ang-2 protein in tumor samples. For the validation of the antibody used, please see Answer number 3, the second chapter. 

Added negatively, in Abstract, page 2, line 57.

5. Line 235: the authors state that “the correlation [between Ang-2 and Tie-1] was significant among other histological types”. If so, please present r and p values in the text or a table. Please clarify in the Methods section which statistical test was used to measure this correlation.

Added r and p values in the text. Results, page 13, lines 246.

Correlations between immunohistochemical parameters were measured using the Spearman`s correlation test. Methods, page 10, lines 197-98.

6. In Tables 2 and 3, the authors seem to only present data for analyses that yielded statistically significant results. Did all analyses performed yield statistically significant results? Please provide the results of all analyses for completeness. Were all parameters analysed for their relationship to overall survival also assessed regarding PFS? If so, please present these data as well.

All analyses performed did not yield statistically significant results. All parameters analysed for their relationship to overall survival were also assessed regarding PFS. 

Non-significant and PFS results are added to the Table 3 and Table 4 (former Table 2 and 3), pages 16-18. 

7. Similarly, if correlation analyses between IRS and qRTPCR results were performed for Tie1 and Tie2, please present these as well. Given that the authors have removed most of the qRTPCR data from the previous version of their manuscript, they should consider toning down claims of their study’s importance/uniqueness related to this.

Correlation analyses between IRS and qRTPCR results were also performed for Tie-1 and Tie-2. These results are added to the Supplementary Table 3 (S3 Table). 

In the first revision, we were suggested to remove the qRTPCR data. As a compromise, we left it to the Supplementary material, because on the other hand it was considered the base of the result. 

8. The authors state (line 97) that “there are no studies investigating the tissue expressions of Ang-2 and its receptors in ovarian cancer”, referencing Table S1, yet this Table contains several examples where such investigations appear to have been published, e.g. ref. 19. Please revise this statement.

Revised the statement, ”there are only four studies with smaller populations and none investigating the tissue expression of Ang-2 and both its receptors in ovarian cancer”. Introduction, page 4, lines 97-98.

9. Table S1: several references within this table do not appear within the main References list. Please add a list of full Supplementary references to the Supporting information file to cover these.

List of full Supplementary references is added in Supplementary material.

10. Statistical analysis: I am not familiar with the use of a Mann-Whitney U test as a post-hoc test for pairwise comparisons subsequent to the overarching Kruskal-Wallis test. In my understanding Dunn’s post-hoc test is more appropriate. Please provide a reference validating the appropriateness of this approach, or seek expert statistical advice.

We have seeked expert statistical advice and according to the expert, the results are the same either by using the Mann-Whitney U test or Dunn`s post-hoc test. Both tests are appropriate and equal in this purpose, used as a post-hoc test for pairwise comparisons subsequent to the overarching Kruskal-Wallis test. 

11. Do the omental metastases examined reflect systemic or local/in transit metastasis?

Omental metastases reflect local/ in transit intra-abdominal metastasis in ovarian cancer. It is the most typical intra-abdominal site for ovarian cancer to metastasize and omental metastasis over 2cm makes the ovarian cancer disseminated stage IIIc. 

12. S1 Table 1st row data, 3rd column: “Post-oper” – word incomplete.

Corrected, Supplementary S1 Table, 1st row. 

13. Line 326: “angiogenetic” should be “angiogenic”

Corrected, page 19, line 342.

Response to Reviewer 4

We thank Reviewer for the valuable comments. Each of your comment or question is indicated in bold followed by our response. All changes are also highlighted in the “Revised manuscript with track changes” and corrected in the unmarked version of revised manuscript without track changes.

1. Figure 1 – Figure 1 and the Figure 1 legend both lack sufficient detail to interpret the immunohistochemistry data. Readers should be able to determine the distribution of staining scores across the cohort, which is not possible with data presented as a histogram. Please change histograms to violin plots, which will allow you to show the median score in each group, the individual values per patient, their distribution and the density. The figure or legend must also state the number of patients included in each group. A statistical test should also be used to compare expression values for each marker between primary tumors and metastases, and P values should be reported for each comparison. The y-axis also should be labelled. Please also state in the legend whether the IRS was calculated from the tumor epithelium or stroma.

Figure 1 is now changed to violin plots. The y-axis is labelled as IRS and significant p-values are reported for each comparison in Figure 1.

The number of patients included in each group, statistical test and from which tissue/ compartment IRS was calculated were added to the Figure legend.

Results, page 10-11, lines 211-18. 

2. Results – Throughout the text the authors refer to staining patterns in primary high grade serous and endometrioid cancer. This data is not shown. For example, lines 205-206 – The authors report that “the stromal staining of Ang-2 was mostly weak in 74% of cases in primary high grade serous and endometrioid tumors”. No Figure or Table is referred to. When referring to staining intensity, perhaps the authors mean to refer to Table S2? However, Table S2 contains staining intensity for all histotypes combined. Where is the data presented for selected primary high grade serous and endometrioid tumors?

To make the article more readable and not to repeat all the results, we decided to write the results of high grade serous and endometrioid cancer in the main text and the whole study population in the Table 2 (former Table S2). High grade serous cancer is the most common type of ovarian cancer and the results of that particular group are of the most interest. 

Clarified in Results page 11, rows 237-38.

3. Results, lines 224-225 – “stromal staining was strong in 64% of cases, emphasizing stromal fibroblasts and vascular endothelium (Figs 1 and 2c-d)”. Figures 1 and 2 do not show that 64% of cases have strong stromal staining. Authors need to carefully check that each results statement is correctly linked to the appropriate Figure and/or Table, and that the Figures and/or Tables referred to actually contain that data.

Corrected the place of reference to the ”Figs 1 and 2c-d”.

Results, page 11, line 228.

Checked the references to the Figures and Tables.

4. Table S2 – Given the amount of key data presented in Table S2, this should be changed to a main Table within the manuscript.

Table S2 is relocated and changed to a main table within the manuscript, Table 2, pages 12-13. 

5. Table S2 – It is not clear whether the P values are calculated for just the paired primary and metastatic samples from the same patients, or unpaired groups of all primary and all metastases. Consider doing both. The range of IRS values are missing from the table.

P values are calculated for the paired or related primary and metastatic samples from the same patients. That is clarified in the Table 2 (former S2 Table).

We do not want to mix all the different histological types and compare them to the metastasis of high grade serous tumors, as all the metastasis were high grade serous type. However there was not significant differencies to the results presented in the manuscript even when calculated with the unpaired groups. 

The ranges of IRS are added to the Table 2 (former S2 Table). 

6. Results, lines 234-237 – “The level of Tie-1 expression was moderately, but significantly, correlated with its ligand Ang-2 (r = 0.5, p < 0.001) in primary serous ovarian tumors, and the correlation was also significant among other histological types. The expression level of Tie-2 showed a weak, but significant correlation with Ang-2 (r = 0.3, p = 0.012) and moderate correlation with Tie-1 (r = 0.5, p < 0.001) in primary cancer.” Which values were used to test these correlations and in which cellular compartment?

IRS and PP values were used to test these correlations. IRS and PP are measured from the expressions of the tumor epithelial cells. 

Clarified in Results page 13, lines 244-50.

7. Results, lines 246-247 – “The level of Tie-1 expression did not differ between primary and metastatic lesions (Fig 1, 2 and S2 Table)”. However table S2 shows that the stromal expression of Tie-2 does differ between primary and metastatic lesions (p-value 0.046. Please clarify.

Clarified in Results, page 14, lines 259-61.

8. Table S3 – Please indicate whether the values were calculated from epithelial or stromal cells.

The values were calculated from the tumor epithelial cells. 

Added in Supporting information captions, page 28, lines 609-10.

9. Results, lines 251-270 – Throughout this section the authors are investigating the association between biomarker expression and various clinical parameters. However, we are not shown the expression data. For example, “The low level epithelial expression of Ang-2 in primary tumors associated significantly with the ovarian cancer recurrence and with the greater residual tumor size (� 1cm) after primary surgery (p = 0.018, p = 0.012)”. For this comparison, the authors should show the median IRS for tumours with residual tumor > 1cm, compared to IRS of tumours with residual tumor < 1cm and no residual tumour. Therefore, in addition to the P values in Table S3, please include the median IRS and PP values for each group comparison.

Table S2 (former Table S3) is made newly and now it shows median IRS an PP values for each group comparison. Checked by the statistician, in non-normally distributed groups, the median value can be the same even if there is a significant difference between the groups. 

10. Results, lines 254-256 – “In high grade serous tumors, high levels of Ang-2 staining were related to the resistance to platinum chemotherapy (p = 0.017) (S3 Table).” Platinum resistance and high-grade serous tumors are both not shown in Table S3.

Platinum resistance and high grade serous cancer are mentioned only in the text, because other groups and the total study population had not significant results in relation to that parameter. S2 Table (former S3 Table) mainly consists of the results of the total study population although some results of the HGSC are shown for the interest of that particular group, as being the most common type of ovarian cancer. 

11. Results, line 284-286 – The authors need to explain how high Tie-2 is associated with shorter overall survival, and low Ang-2 is associated with poor survival, and yet Tie-2 and Ang-2 expression are positively correlated (as stated in Results lines 236-237). Presumably these are different populations of patients? Please explore and explain this result.

IRS >2 of Tie-2 is associated with shorter overall survival of high grade serous ovarian cancer patients as indicated in Table 4. In turn IRS of Ang-2 ≤6 is related to poor survival when the whole study population is taken into count. This is clarified in Table 4, page 18.

When the whole study population is observed, Tie-2 and Ang-2 expressions are positively correlated. The other histologies in addition to high garde serous tumors make the correlation positive. In addition, it has to be noticed that the other one is the receptor while the other is the actual factor. 

12. Figure 3 – In the figure, please indicate the number of individuals at risk at each major time interval under each Kaplan-Meier plot.

Indicated the number of individuals at risk at each major time intervals under each Kaplan-Meier plot in Figure 3. 

13. Figure 3 and Table 3 – How was high and low assigned for each biomarker, and what was the cut-off used for high and low?

The median of the group was used for the cut-off for high and low for each biomarker. 

14. qRT-PCR is mentioned in the Abstract, the Methods and Discussion, however not mentioned in the Results. Please clarify.

The results of qRT-PCR were relocated to the Supporting information (S3 Table) after previous revision. In the first revision, we were suggested to remove the qRTPCR data. As a compromise, we left it to the Supplementary material, because on the other hand it was considered the base of the result. 

15. Table S4 – Please include a table description indicating that Ang-2 IRS and PP is being correlated with qRT-PCR, and how the P values were calculated, and the numbers of samples tested. Also, please include the data for Tie-1 and Tie-2.

Table description is included with the information mentioned above. Supporting information captions S3 Table (former S4 Table) page 28, lines 612-14.

qRT-PCR data for Tie-1 and Tie-2 and statistical test is included in S3 Table. 

16. Grammar needs to be corrected throughout the manuscript.

Prof Ewen MacDonald corrected the english grammar of the manuscript.

17. Table 3 – Table 3 shows 48 patients with low Ang-2 IRS and 82 patients with high Ang-2, which added together equals 130 patients. However, there are only 86 women described in the Materials and Methods. Please clarify. Also, Tie-1 is missing from this Table.

Table 4 (former Table 3) and that particular column shows the 5 year survival percentage of each group mentioned, not the absolute number of patients. So, 48% of patients with low Ang-2 IRS are alive after five years of diagnosis and further, 82% of patients with high Ang-2 are alive after five years. 

Tie-1 data is added to the Table 4, page 18.

---

## [Decision Letter · Decision Letter 2]

24 Jun 2020

PONE-D-19-33270R2

HIGH EXPRESSION OF TIE-2 PREDICTS POOR PROGNOSIS IN PRIMARY HIGH GRADE SEROUS OVARIAN CANCER

PLOS ONE

Dear Dr. Sopo,

Thank you for submitting your manuscript to PLOS ONE. After careful consideration, we feel that it has merit but does not fully meet PLOS ONE’s publication criteria as it currently stands. Therefore, we invite you to submit a revised version of the manuscript that addresses all of the points raised during the review process by both reviewers.

We look forward to receiving your revised manuscript.

Kind regards,

Elizabeth Christie

Academic Editor

PLOS ONE

Reviewers' comments:

Reviewer's Responses to Questions

**Comments to the Author**

1. If the authors have adequately addressed your comments raised in a previous round of review and you feel that this manuscript is now acceptable for publication, you may indicate that here to bypass the “Comments to the Author” section, enter your conflict of interest statement in the “Confidential to Editor” section, and submit your "Accept" recommendation.

Reviewer #3: (No Response)

Reviewer #4: (No Response)

2. Is the manuscript technically sound, and do the data support the conclusions?

Reviewer #3: Yes

Reviewer #4: No

3. Has the statistical analysis been performed appropriately and rigorously? 

Reviewer #3: Yes

Reviewer #4: No

4. Have the authors made all data underlying the findings in their manuscript fully available?

Reviewer #3: No

Reviewer #4: No

5. Is the manuscript presented in an intelligible fashion and written in standard English?

Reviewer #3: Yes

Reviewer #4: No

6. Review Comments to the Author

Reviewer #3: The manuscript is overall improved, but there are some important points remaining to be addressed. Please see below, with numbers corresponding to the original questions raised. All points not specifically mentioned have been satisfactorily addressed.

1. The authors have acknowledged the potential role of Ang2 in EMT and metastasis, but have done so very superficially – in fact the statement I made regarding this in my initial review has been copied almost word for word into the manuscript, which is not acceptable. Please add your own insight. In asking for the authors to consider the limitations of the conclusions that can be drawn from their data re angiogenesis, I intended a more balanced assessment of the different possible mechanisms at play, rather than the blunt statement that has been inserted. Although the Discussion is overall appropriate, it remains unclear to me whether the observed relationship between tumour cell-expressed Tie2/Ang2 and survival/metastasis has anything to do with angiogenesis, and if so, how. A correlation analysis between tumour cell-expressed Tie2/Ang2 and blood vessel density would have been informative in this regard, but perhaps this must wait for a future study. See also point 11.

3. Thankyou for adding the images in S1 Figure. Please add detail in the caption to describe what tissues are stained in each image, and what the negative control represents – negative control tissue, or negative control antibody?

4. These results are still somewhat perplexing, but they appear to be experimentally sound and sufficiently explained.

6. Tables 3 and 4 still have data missing. Please present all multivariate hazard ratios, confidence intervals and p values for all variables assessed in Table 3, and all HRs, CIs and p values in Table 4. There should be no blank spaces in the tables – all data should be presented regardless of significance so readers can transparently assess the results. Statistically significant results can be highlighted with asterisks or bolding of numbers if desired – if so please make sure this is done consistently across all tables.

7. Thankyou for presenting complete data in Table S3. However there still appears to be data missing from Table S2. Please fill in the blanks as in point 6. Also, in table 2, some entries contain two numbers separated by a comma. What is meant by this? Should the comma be a decimal point?

8. Please cite the four studies in the text on p4.

9. Please ensure all references in the Supplementary materials and the main manuscript are formatted the same way, consistent with PLoS One’s standards. Some references have authors’ first names while most have only initials.

11. The Discussion regarding omental metastases (e.g. p21 lines 389-406) could benefit from more detail as to how omental metastases occur, if this is known. Do these secondary tumours form as in-transit metastases along blood vessels or lymphatics? Or are they simply the result of local invasion through the parenchyme or along the surface of the omentum, only engaging blood vessels when the deposit becomes large enough to be hypoxic? This also relates back to point 1.

Reviewer #4: 1) Figure 1 – The median is difficult to see when represented as a larger dot of the same colour as all the other data points. Please use a horizontal line of a different colour to indicate the median IRS. Also, please use grey transparent dots for each data point, to allow readers to distinguish data points that are close together.

2) Throughout the manuscript, the authors refer interchangeably to “high-grade serous ovarian tumors”, and “high grade serous and endometrioid tumors”, and “primary high grade cancer”, and “primary ovarian tumors”. This makes the text difficult to follow. According to Table 1, high-grade serous tumors make up 45 of the 86 ovarian carcinomas tested in this study. If the authors would like to draw specific conclusions about high-grade serous (e.g. lines 201-202), please refer to figures or tables that display results for those specific 45 high-grade serous tumors. Or equally, if the authors would like to make conclusions about high-grade serous and high-grade endometrioid combined (e.g. lines 205-207), please refer to figures or tables in which those 45 high-grade serous have been combined with the 13 high-grade endometrioid tumors, and provide a rationale for doing so. Or if the authors would like to present results pertaining to all ovarian carcinomas combined (e.g. Figure 1 and Table 2), please clearly state this in the text.

3) Related to the above issue, throughout the text, many statements in the Results section are still not linked to a Figure or Table. There are too many to list.

4) According to the Methods, ten randomly selected microscopic fields were examined per tumor section. Please include in the Methods a statistical measure on how consistent the scores were between microscopic fields, and state whether the final scores for each tumor were an average of all 10 fields.

5) Table 2 – How the p values relate to the data presented in the table is still unclear. If the paired Wilcoxon test was used to compare expression of the paired primary and metastatic high-grade serous tumours (as stated by the authors response to reviewer), the table should show the IRS, PP and SI results for the 16 primary high-grade serous tumours alongside their matched 16 high-grade serous metastases, as that is the data that the Wilcoxon test is being applied to. In addition, all the other unpaired primary tumour IRS, PP and SI results should also be presented in a separate column. Also, Wilcoxon seems appropriate to compare paired IRS and PP scores, but how would that be applied to SI? The SI results are presented in the table as a proportion of tumours that are weak, moderate or strong, so wouldn’t a chi-square test be more appropriate?

6) Table S2 – This table is hard to follow as currently presented. Why is SI reported under Residual tumor, and not the other clinical characteristics, and how is SI reported here as values between 62 and 88, when in the methods it was scored as “0, negative; 1, weak; 2, moderate; 3, strong”? In addition, for some clinical characteristics there appear to be too many p values reported, whilst under Histology the PP Tie-1 p value is missing. Please edit this table to be easier to follow.

7) Results, lines 266-268, the authors state that “The low epithelial expression of Ang-2 in primary tumors associated significantly with the ovarian cancer recurrence and with the greater residual tumor size (� 1cm) after primary surgery (p = 0.018, p = 0.012).” However, in Table S2 the median IRS of Ang-2 is 6 both in patients with recurrence and without recurrence, and the median IRS of Ang-2 is also 6 regardless of residual tumor status, which seems to indicate there is no significant difference between any of these groups. This calls into question the conclusions drawn from this part of the analysis.

8) Figure 3 – Kaplan Meier curves are shown for HGSC patients for Tie-2 (n=45), but not for all ovarian (n=86). Whereas K-M curves are shown for all patients for Ang-2 (n=86), but not HGSC (n=45). Please show the omitted Kaplan-Meier curves.

9) If the authors have noted in their correspondence to reviewers that Tie-2 and Ang-2 are correlated in non-HGSC tumours, and do not correlate in HGSC, please state this explicitly in the text of the manuscript for readers, and use the Discussion to explore what this might mean.

10) Please state in the Methods how the high and low cut-offs were assigned for each biomarker.

11) Table S3 - The qRT-PCR data needs to be mentioned in the Results, even if the Table was relocated to the Supporting Information. This data does call into question the specificity of the antibodies, or the specificity of the PCR, considering the Ang-2 correlation was significantly negatively correlated, and the Tie-1 and Tie-2 correlations were considerably weak.

7. PLOS authors have the option to publish the peer review history of their article (what does this mean?). If published, this will include your full peer review and any attached files.

Reviewer #3: No

Reviewer #4: No

---

## [Author Response · Author response to Decision Letter 2]

17 Aug 2020

Response to Reviewer 3

We thank Reviewer for the valuable comments. Each of your comment or question is indicated in bold followed by our response. All changes are also highlighted in the “Revised manuscript with track changes” and corrected in the unmarked version of revised manuscript without track changes.

Reviewer #3: The manuscript is overall improved, but there are some important points remaining to be addressed. Please see below, with numbers corresponding to the original questions raised. All points not specifically mentioned have been satisfactorily addressed.

1. The authors have acknowledged the potential role of Ang2 in EMT and metastasis, but have done so very superficially – in fact the statement I made regarding this in my initial review has been copied almost word for word into the manuscript, which is not acceptable. Please add your own insight. In asking for the authors to consider the limitations of the conclusions that can be drawn from their data re angiogenesis, I intended a more balanced assessment of the different possible mechanisms at play, rather than the blunt statement that has been inserted. Although the Discussion is overall appropriate, it remains unclear to me whether the observed relationship between tumour cell-expressed Tie2/Ang2 and survival/metastasis has anything to do with angiogenesis, and if so, how. A correlation analysis between tumour cell-expressed Tie2/Ang2 and blood vessel density would have been informative in this regard, but perhaps this must wait for a future study. See also point 11.

We have added our insight of the potential role of Ang-2 in EMT and metastasis and also the assessement of the different possible mechanisms in metastasis in Discussion, page 22, rows 411-423.

In earlier studies, it has been concluded that tumor cells express and secrete several angiogenic factors including Tie2/Ang2 to the tumor microenvironment (for ex. Gavalas N, Int J Mol Sci 2013).

As reviewer mentions, the correlation of Tie-2/Ang-2 with blood vessel density will need further study.

2. Original question 3. Thank you for adding the images in S1 Figure. Please add detail in the caption to describe what tissues are stained in each image, and what the negative control represents – negative control tissue, or negative control antibody?

Details of the caption for S1 Figure added, page 29, rows 643-646.

A sample without a primary antibody was used as a negative control, added in Methods, page 8, rows 155-156.

3. Original question 4. These results are still somewhat perplexing, but they appear to be experimentally sound and sufficiently explained.

We don`t have anything else to add.

4. Original question 6. Tables 3 and 4 still have data missing. Please present all multivariate hazard ratios, confidence intervals and p values for all variables assessed in Table 3, and all HRs, CIs and p values in Table 4. There should be no blank spaces in the tables – all data should be presented regardless of significance so readers can transparently assess the results. Statistically significant results can be highlighted with asterisks or bolding of numbers if desired – if so please make sure this is done consistently across all tables.

All assessed multivariate hazard ratios, confidence intervals and p values are added in Table 3 and 4. Pages 17-18.

Statistically significant p-values are bolded in all Tables. 

5. Original question 7. Thank you for presenting complete data in Table S3. However there still appears to be data missing from Table S2. Please fill in the blanks as in point 6. Also, in table 2, some entries contain two numbers separated by a comma. What is meant by this? Should the comma be a decimal point?

P-values added in former Table S2, now named Table S3 (changed according to the reference order in the manuscript). 

Commas of the entries changed to the decimal points in Table 2, page 12-13. 

6. Original question 8. Please cite the four studies in the text on p4.

Four studies cited in the manuscript, page 4, row 98. References 12-15.

7. Original question 9. Please ensure all references in the Supplementary materials and the main manuscript are formatted the same way, consistent with PLoS One’s standards. Some references have authors’ first names while most have only initials.

References of the manuscript and the Supplementary material are corrected and formatted consistently with PLos One`s standards. Authors first names have removed and changed to initials. 

8. Original question 11. The Discussion regarding omental metastases (e.g. p21 lines 389-406) could benefit from more detail as to how omental metastases occur, if this is known. Do these secondary tumours form as in-transit metastases along blood vessels or lymphatics? Or are they simply the result of local invasion through the parenchyme or along the surface of the omentum, only engaging blood vessels when the deposit becomes large enough to be hypoxic? This also relates back to point 1.

We have added the principle mechanism of omental metastasis in Discussion, page 22, rows 414-417. 

The omental metastasis is the result of implantation of the cancer cells, exfoliated from the primary tumors and spreaded by ascites, to the omental surface and engaging blood vessels when the tumor becomes large enough to be hypoxic.

Response to Reviewer 4

We thank Reviewer for the valuable comments. Each of your comment or question is indicated in bold followed by our response. All changes are also highlighted in the “Revised manuscript with track changes” and corrected in the unmarked version of revised manuscript without track changes.

1. Figure 1 – The median is difficult to see when represented as a larger dot of the same colour as all the other data points. Please use a horizontal line of a different colour to indicate the median IRS. Also, please use grey transparent dots for each data point, to allow readers to distinguish data points that are close together.

Each single data point is now changed to grey dots to distinguish them from each other and from the black dot describing the median values of the groups in Figure 1. 

2. Throughout the manuscript, the authors refer interchangeably to “high-grade serous ovarian tumors”, and “high grade serous and endometrioid tumors”, and “primary high grade cancer”, and “primary ovarian tumors”. This makes the text difficult to follow. According to Table 1, high-grade serous tumors make up 45 of the 86 ovarian carcinomas tested in this study. If the authors would like to draw specific conclusions about high-grade serous (e.g. lines 201-202), please refer to figures or tables that display results for those specific 45 high-grade serous tumors. Or equally, if the authors would like to make conclusions about high-grade serous and high-grade endometrioid combined (e.g. lines 205-207), please refer to figures or tables in which those 45 high-grade serous have been combined with the 13 high-grade endometrioid tumors, and provide a rationale for doing so. Or if the authors would like to present results pertaining to all ovarian carcinomas combined (e.g. Figure 1 and Table 2), please clearly state this in the text.

We have standardized the reference of the groups in the manuscript by removing ”high grade serous” and ”high grade serous and endometrioid” groups from the text and pertaining all primary ovarian tumors and refering to the Figure 1 and Table 2. 

Pages 10-11, rows 205, 208-209, 232, 236

When comparing related primary and metastatic tumors, the term ”high grade serous” is used, because all 16 metastases were high grade serous as well as obviously their primary tumors. Also the clearest advantage of overall survival was seen in high grade serous group, so that`s why in Table 3 and 4, the high grade serous group is reported separately. 

3. Related to the above issue, throughout the text, many statements in the Results section are still not linked to a Figure or Table. There are too many to list.

We have added the references to the Tables and Figures. Page 10, rows 208, 210, 212; page 11, rows 231, 233, 235, 237; page 14, rows 260, 268, 269; page 15, rows 278, 285, 288, 292; page 16, rows 303, 305, 306, 310.

To make it more convenient for the readers, we would not want to repeat the references to the Tables and Figures after every sentence. 

4. According to the Methods, ten randomly selected microscopic fields were examined per tumor section. Please include in the Methods a statistical measure on how consistent the scores were between microscopic fields, and state whether the final scores for each tumor were an average of all 10 fields.

It is not possible to present the numerical values of each single microscopic fields as there were 306 samples with ten microscopic fields each making 3060 calculations. The mean percentage of positive stained cells and intensity was estimated as stated in the Methods, page 8, rows 159-166. 

5. Table 2 – How the p values relate to the data presented in the table is still unclear. If the paired Wilcoxon test was used to compare expression of the paired primary and metastatic high-grade serous tumours (as stated by the authors response to reviewer), the table should show the IRS, PP and SI results for the 16 primary high-grade serous tumours alongside their matched 16 high-grade serous metastases, as that is the data that the Wilcoxon test is being applied to. In addition, all the other unpaired primary tumour IRS, PP and SI results should also be presented in a separate column. Also, Wilcoxon seems appropriate to compare paired IRS and PP scores, but how would that be applied to SI? The SI results are presented in the table as a proportion of tumours that are weak, moderate or strong, so wouldn’t a chi-square test be more appropriate?

The results of the 16 primary high grade serous tumors are now added to the Table 2 alongside with their matched metastases. The results of the whole study population are left in their own column. Wilcoxon test is used to compare paired IRS and PP values and Pearson`s chi-square test to SI results and that is added to the Table 2 caption, page 12-13.

6. Table S2 – This table is hard to follow as currently presented. Why is SI reported under Residual tumor, and not the other clinical characteristics, and how is SI reported here as values between 62 and 88, when in the methods it was scored as “0, negative; 1, weak; 2, moderate; 3, strong”? In addition, for some clinical characteristics there appear to be too many p values reported, whilst under Histology the PP Tie-1 p value is missing. Please edit this table to be easier to follow.

We modified the former Table S2 (now Table S3) according to the suggestions of the previous revision and added the median values of each subgroup in the Table. Now, to make it more readable we took the SI of the residual tumor group away (there were the percentage of the strong staining values reported as 62, 86 and 88, not the exact scores). 

Also according to the suggestions of Reviewer 3, all the p-values should be presented, so we added all the p-values, also the values under Histology (and PP Tie-1) groups. 

7. Results, lines 266-268, the authors state that “The low epithelial expression of Ang-2 in primary tumors associated significantly with the ovarian cancer recurrence and with the greater residual tumor size (� 1cm) after primary surgery (p = 0.018, p = 0.012).” However, in Table S2 the median IRS of Ang-2 is 6 both in patients with recurrence and without recurrence, and the median IRS of Ang-2 is also 6 regardless of residual tumor status, which seems to indicate there is no significant difference between any of these groups. This calls into question the conclusions drawn from this part of the analysis.

We were also wondering that, but as I mentioned in the last revision, according to the statistician, in non-normally distributed groups, the median value can be the same even if there is a significant difference between the groups. 

I also calculated this by checking the IRS values of each subgroup and its individual patients. For example in ”no recurrence” group there were 17 patients of total 27 patients (37%), who had low Ang-2 IRS (6 or under) and in ”yes recurrence” group there were 35 patients of total 40 (88%), who had low Ang-2 IRS. And yet the median IRS of both groups was 6. Similarly with the primary recidual tumor, in the ”no recidual” group 28/40 patients (70%) had low Ang-2 IRS and in ”recidual >1cm” group 37/40 (93%) had low Ang-2 IRS. 

8. Figure 3 – Kaplan Meier curves are shown for HGSC patients for Tie-2 (n=45), but not for all ovarian (n=86). Whereas K-M curves are shown for all patients for Ang-2 (n=86), but not HGSC (n=45). Please show the omitted Kaplan-Meier curves.

Kaplan Meier curves for Tie-2 IRS of all ovarian carcinoma patients and Ang-2 of HGSC group are added to the Figure 3 and Figure caption, page 16, rows 323-324.

9. If the authors have noted in their correspondence to reviewers that Tie-2 and Ang-2 are correlated in non-HGSC tumours, and do not correlate in HGSC, please state this explicitly in the text of the manuscript for readers, and use the Discussion to explore what this might mean.

As we stated in the manuscript Tie-2 PP and Ang-2 PP of the whole study population had a weak positive correlation, but corresponding IRSs of the whole group did not reach the significance. When considering the high grade serous group neither IRS or PP of Tie-2 and Ang-2 had significant correlation. This might support the theory of high Ang-2 not being a significant factor for poor OS as we have mentioned in Discussion, page 23, rows 438-439. Otherwise prolonging the Discussion further, as IRS is the main parameter to make conclusions, is not relevant and affects the readability of the manuscript in our opinion. This is also because of the very weak correlation of only PPs.

Added in the Results, page 13, row 251. 

10. Please state in the Methods how the high and low cut-offs were assigned for each biomarker.

Assignement of cut-off values for each biomarker was made according to median value of the biomarker and this is now stated in Methods, page 8-9, rows 166-169.

11. Table S3 - The qRT-PCR data needs to be mentioned in the Results, even if the Table was relocated to the Supporting Information. This data does call into question the specificity of the antibodies, or the specificity of the PCR, considering the Ang-2 correlation was significantly negatively correlated, and the Tie-1 and Tie-2 correlations were considerably weak.

Chapter of qRT-PCR data added to the Results, page 14, rows 265-272. 

As mentioned previously the expression of tumor cells was evaluated by immunohistochemistry in this study and PCR method is not able to evaluate the localization of the biomarker. In usual qRT-PCR analysis, different tissues including tumor stroma are mixed.

---

## [Decision Letter · Decision Letter 3]

3 Sep 2020

PONE-D-19-33270R3

HIGH EXPRESSION OF TIE-2 PREDICTS POOR PROGNOSIS IN PRIMARY HIGH GRADE SEROUS OVARIAN CANCER

PLOS ONE

Dear Dr. Sopo,

Thank you for submitting your manuscript to PLOS ONE. After careful consideration, we feel that it has merit but does not fully meet PLOS ONE’s publication criteria as it currently stands. Therefore, we invite you to submit a revised version of the manuscript that addresses the points raised during the review process.

Please address the comments made by the reviewers. Additionally, please note that Reviewer #4 has said that not all data underlying the findings in the manuscript are fully available (see point 4 below) - please ensure this is addressed in your response.

Additionally, Reviewer #3 has noted a number of typographical and grammatical errors. *PLOS ONE* does not copyedit accepted manuscripts, so the language in submitted articles must be clear, correct, and unambiguous, therefore we suggest you thoroughly copyedit your manuscript for language usage, spelling and grammar. If you do not know anyone who can help you with this, you may wish to consider employing a professional scientific editing service. 

We look forward to receiving your revised manuscript.

Kind regards,

Elizabeth Christie

Academic Editor

PLOS ONE

Reviewers' comments:

Reviewer's Responses to Questions

**Comments to the Author**

1. If the authors have adequately addressed your comments raised in a previous round of review and you feel that this manuscript is now acceptable for publication, you may indicate that here to bypass the “Comments to the Author” section, enter your conflict of interest statement in the “Confidential to Editor” section, and submit your "Accept" recommendation.

Reviewer #3: (No Response)

Reviewer #4: All comments have been addressed

2. Is the manuscript technically sound, and do the data support the conclusions?

Reviewer #3: Yes

Reviewer #4: Yes

3. Has the statistical analysis been performed appropriately and rigorously? 

Reviewer #3: Yes

Reviewer #4: Yes

4. Have the authors made all data underlying the findings in their manuscript fully available?

Reviewer #3: Yes

Reviewer #4: No

5. Is the manuscript presented in an intelligible fashion and written in standard English?

Reviewer #3: Yes

Reviewer #4: Yes

6. Review Comments to the Author

Reviewer #3: 1. This section of the Discussion is much improved. However, in acknowledgement that this study does not provide any direct conclusions regarding the relationship between Ang/Tie expression and angiogenesis in ovarian cancer, please delete the word “angiogenic” from the conclusion in line 452, and insert the word “potentially” before “reflecting the more active angiogenesis” on line 449. Possible effects of tumour-expressed Tie2 and Ang2 on pro-angiogenic signalling through endothelial-expressed receptors can be inferred and discussed but they have not been shown directly in this study.

2. Original Q3: Thankyou for the additional detail in the caption for Figure S1, but please be more precise with the descriptions. In the Methods the authors have defined the negative controls as tissues stained with no primary antibody, whereas in the caption the negative controls are described as “negative control tissue”. “Negative control tissue” would imply an image derived by staining a tissue that does not endogenously express a protein with a specific antibody for that protein. Panels b), d) and f) are probably better described as “ovarian carcinoma negative control staining (primary antibody omitted)”. A more stringent standard for IHC controls would be to perform specific antibody staining and negative control staining on serial sections of the same (positively staining) tissue; thus it is not ideal that the Tie1 positive control is tonsil and the negative control is ovarian carcinoma.

3. Original Q4: The fact that IHC and qRT-PCR results for Ang2 correlate negatively in one tissue and positively in another is confusing. The authors may therefore wish to consider removing this information from the abstract.

4. Original Q6: It will be important for the authors to check the final formatting of Table 3 by the journal to ensure the results are read correctly and the results and parameter headings are in the correct hierarchy. I would suggest adding a subheading “immunohistochemical staining” above the IRS results and e.g. “clinical features” above stage/ascites and then un-bolding the IRS parameters and stage/ascites to bring these into better alignment with the other sections. This will make it clearer which comparisons relate to overall survival and which to progression free survival when both sets of results are presented within the one table.

5. Original Q7: The commas in Tables 2 and S3 appear not to have been changed to decimal points. Please amend. Also the presentation of the data in Table S3 is still confusing. Please clarify in the table caption if the data presented here represent epithelial staining, stromal staining or both. The text (p15 line 285) states that low Tie-1 expression was associated with a greater residual tumour size (>1cm) after primary surgery (p = 0.008), but in Table S3 the p value 0.008 is placed corresponding to zero residual tumour. Similarly I cannot find the stated p = 0.006 corresponding to significantly lower Tie1 expression in high grade tumours, and there are several other examples where the p values cited in the text cannot be found against the corresponding variables in Table S3. Why is this? I assume the remaining blank spaces against a parameter in this table (e.g. for >1 cm residual tumour, Yes recurrence) imply that that parameter was used as the reference for the comparison (with the exception of Histology where there is no obvious reference). Should the 0 residual tumour be the reference condition in Table S3 as it is in Table 3? Please confirm this and ensure that the results are correctly aligned within the table. Also change “0” residual tumour in Table S3 to “None” to be consistent with Table 3. In Table 4 the reference conditions are explicitly labelled but I don’t think this is the norm. Be consistent throughout all tables.

6. Original Q8: OK

7. Original Q9: OK

8. Original Q11: Thankyou, this part of the discussion is much improved.

Other comments and Typographical/grammatical errors:

I have roughly proofread the manuscript for English expression. Note that this journal does not provide copyediting, so the authors must please proofread all text, figures and tables very carefully.

Figures and Tables should be referred to as “Figure X” or “Table X” throughout – this appears to only be the wrong way around for the supplementary figures and tables.

P2 line 37: add comma after “ovarian cancer”

P2 line 39 and line 41: change “expressions” to “expression”, and delete “the” before “angiopoietin-2” on line 39.

P3 line 64: add comma after “angiogenesis”

P3 line 70: There are many endothelial growth factor systems. Sentence should read “Other endothelial growth factor systems, such as the angiopoietin-Tie complex, have not been as extensively studied.”

P3 line 81: replace “speeded up” with “accelerated”; remove “the” before “targeted”

P4 lines 89-90: amend to read “has prolonged median progression free survival (PFS)”

P4 line 100: delete “the” before “dissemination”

P4 lines 103, 104: amend “expressions” to “expression

P8 line 143: delete “the” before “written”

p11 line 238: amend to “between expression in”

p14 lines 255-257: section heading could be amended to “Expression of angiogenic factors in primary high grade serous tumours as compared to related metastases”. First sentence could be amended to “Expression of both Ang-2 and Tie-2 …. was”

p14 lines 265-272: the concept of “qRT-PCR levels” doesn’t make sense – qRT-PCR is a technique that measures mRNA levels. I would suggest amending this section to the following:

Correlation of qRT-PCR results with immunohistochemical staining

Ang-2 IRS was significantly negatively correlated to the mRNA levels of Ang-2 measured by qRT-PCR in primary ovarian cancer (r = -0.64, p = 0.002). The correlation was even stronger when only high grade serous tumors were counted (r = -0.868, p < 0.001) (Table S2). In metastatic tumors Ang-2 PP was strongly correlated to the corresponding mRNA levels (r = 0.752, p = 0.005) (Table S2). Tie-1 and Tie-2 qRT-PCR values correlated strongly to each other (r = 0.919, p < 0.001), but did not have statistically significant correlations to the respective immunohistochemical expression. In metastases, mRNA levels of Ang-2 correlated strongly with Tie-2 mRNA (r= 0.61, p= 0.016).

Likewise in the caption:

S2 Table.

Correlation of qRT-PCR results with immunohistochemical staining. Ang-2 IRS and PP were correlated with relative mRNA levels as measured by qRT-PCR. There were 22 primary tumor samples and 15 metastatic samples in each group.

P14 lines 276-277: delete “the” before “ovarian” and “greater”

P15 line 280: delete “the” before “resistance”

P15 line 297: delete “the” before “ovarian”

P16 line 324: delete “the” before “significance”

P19 line 354: amend “expressed widely” to “widely expressed”

P21 line 390: add comma after “in addition”

P21 line 398: amend to “Expression of Ang-2 and Tie-2”

P22 line 414-415: delete “the” before “omental”; amend “exfoliating” to “exfoliation”; add “the” before “omental surface”

P22 lines 425-427: delete “the” before “lower” and “larger residual tumour”. Also every instance of “the” could be deleted in the sentence beginning “The expression of Tie-1”.

P23 line 434: amend to “Ang-2 specific andtibody and VEGF blocker aflibercept more strongly reduced tumour growth”

P23 line 453: delete “the” before “OS”.

Reviewer #4: (No Response)

7. PLOS authors have the option to publish the peer review history of their article (what does this mean?). If published, this will include your full peer review and any attached files.

Reviewer #3: No

Reviewer #4: No

---

## [Author Response · Author response to Decision Letter 3]

12 Sep 2020

Response to Reviewer 3

We thank Reviewer for the valuable comments. Each of your comment or question is indicated in bold followed by our response. All changes are also highlighted in the “Revised manuscript with track changes” and corrected in the unmarked version of revised manuscript without track changes.

Reviewer #3: 1. This section of the Discussion is much improved. However, in acknowledgement that this study does not provide any direct conclusions regarding the relationship between Ang/Tie expression and angiogenesis in ovarian cancer, please delete the word “angiogenic” from the conclusion in line 452, and insert the word “potentially” before “reflecting the more active angiogenesis” on line 449. Possible effects of tumour-expressed Tie2 and Ang2 on pro-angiogenic signalling through endothelial-expressed receptors can be inferred and discussed but they have not been shown directly in this study.

Deleted word ”angiogenic” in page 23, row 449.

Inserted word ”potentially” in page 23, row 446.

2. Original Q3: Thank you for the additional detail in the caption for Figure S1, but please be more precise with the descriptions. In the Methods the authors have defined the negative controls as tissues stained with no primary antibody, whereas in the caption the negative controls are described as “negative control tissue”. “Negative control tissue” would imply an image derived by staining a tissue that does not endogenously express a protein with a specific antibody for that protein. Panels b), d) and f) are probably better described as “ovarian carcinoma negative control staining (primary antibody omitted)”. A more stringent standard for IHC controls would be to perform specific antibody staining and negative control staining on serial sections of the same (positively staining) tissue; thus it is not ideal that the Tie1 positive control is tonsil and the negative control is ovarian carcinoma.

Modified the caption for Figure S1, page 28, rows 623-24.

3. Original Q4: The fact that IHC and qRT-PCR results for Ang2 correlate negatively in one tissue and positively in another is confusing. The authors may therefore wish to consider removing this information from the abstract.

Information removed from the Abstract.

4. Original Q6: It will be important for the authors to check the final formatting of Table 3 by the journal to ensure the results are read correctly and the results and parameter headings are in the correct hierarchy. I would suggest adding a subheading “immunohistochemical staining” above the IRS results and e.g. “clinical features” above stage/ascites and then un-bolding the IRS parameters and stage/ascites to bring these into better alignment with the other sections. This will make it clearer which comparisons relate to overall survival and which to progression free survival when both sets of results are presented within the one table.

Subheadings added and parameters unbolded in Table 3, page 17-18.

5. Original Q7: The commas in Tables 2 and S3 appear not to have been changed to decimal points. Please amend. Also the presentation of the data in Table S3 is still confusing. Please clarify in the table caption if the data presented here represent epithelial staining, stromal staining or both. The text (p15 line 285) states that low Tie-1 expression was associated with a greater residual tumour size (>1cm) after primary surgery (p = 0.008), but in Table S3 the p value 0.008 is placed corresponding to zero residual tumour. Similarly I cannot find the stated p = 0.006 corresponding to significantly lower Tie1 expression in high grade tumours, and there are several other examples where the p values cited in the text cannot be found against the corresponding variables in Table S3. Why is this? I assume the remaining blank spaces against a parameter in this table (e.g. for >1 cm residual tumour, Yes recurrence) imply that that parameter was used as the reference for the comparison (with the exception of Histology where there is no obvious reference). Should the 0 residual tumour be the reference condition in Table S3 as it is in Table 3? Please confirm this and ensure that the results are correctly aligned within the table. Also change “0” residual tumour in Table S3 to “None” to be consistent with Table 3. In Table 4 the reference conditions are explicitly labelled but I don’t think this is the norm. Be consistent throughout all tables.

Commas changed to decimal points in Table 2, page 12-13 and S3.

Modified caption of Table S3, page 28, rows 618-19. 

Modified Table S3: 

- P-value 0.008 is placed corresponding zero residual tumor. 

- P-value 0.006 corresponding significantly lower Tie-1 in high grade tumors is associated only with serous* tumors and when the whole study population (all histologies) is taken into account the corresponding p-value is 0.018 (as marked in Table S3 and said later in the manuscript text, page 15, rows 285-86). We prefered to report the results of the whole population. When the result of the whole study population is not significant, the p-value of subgroup of serous histology is reported, if it is significant (marked with *). 

- Consistency with the manuscript text and the Table S3 have been checked. 

- The p-values of the Table S3 are changed to the rows of corresponding significant subgroups the values describe. 

- Yes, the ”0 residual tumor” is the reference group into which the other groups have been compared to and the results are now aligned correctly within the Table S3. 

- The name of ”0 residual tumor” has been changed to ”none residual tumor” in Table S3.

The words ”reference” has been removed in Table 4, page 18. 

6. Original Q8: OK

We do not have anything to add.

7. Original Q9: OK

We do not have anything to add.

8. Original Q11: Thank you, this part of the discussion is much improved.

Thank You for the evaluation and review. 

Other comments and Typographical/grammatical errors:

I have roughly proofread the manuscript for English expression. Note that this journal does not provide copyediting, so the authors must please proofread all text, figures and tables very carefully.

We have proofread all text, figures and tables carefully for English expression.

Figures and Tables should be referred to as “Figure X” or “Table X” throughout – this appears to only be the wrong way around for the supplementary figures and tables.

We have changed the writing form of references for supplementary figures and tables. 

P2 line 37: add comma after “ovarian cancer”

Added comma, page 2, row 37.

P2 line 39 and line 41: change “expressions” to “expression”, and delete “the” before “angiopoietin-2” on line 39.

Modified in page 2, rows 39, 41 and 39.

P3 line 64: add comma after “angiogenesis”

Added comma, page 3, row 61.

P3 line 70: There are many endothelial growth factor systems. Sentence should read “Other endothelial growth factor systems, such as the angiopoietin-Tie complex, have not been as extensively studied.”

Modified, page 3, row 67.

P3 line 81: replace “speeded up” with “accelerated”; remove “the” before “targeted”

Modified, page 3, row 78.

P4 lines 89-90: amend to read “has prolonged median progression free survival (PFS)”

Amended the expression, page 4, rows 86-87.

P4 line 100: delete “the” before “dissemination”

Deleted ”the”, page 4, row 97.

P4 lines 103, 104: amend “expressions” to “expression

Amended, page 4, rows 100, 101.

P8 line 143: delete “the” before “written”

Deleted ”the”, page 7, row 141.

p11 line 238: amend to “between expression in”

Amended, page 11, row 235.

p14 lines 255-257: section heading could be amended to “Expression of angiogenic factors in primary high grade serous tumours as compared to related metastases”. First sentence could be amended to “Expression of both Ang-2 and Tie-2 …. was”

Modified the expression, page 13, rows 252-54.

p14 lines 265-272: the concept of “qRT-PCR levels” doesn’t make sense – qRT-PCR is a technique that measures mRNA levels. I would suggest amending this section to the following:

Correlation of qRT-PCR results with immunohistochemical staining

Ang-2 IRS was significantly negatively correlated to the mRNA levels of Ang-2 measured by qRT-PCR in primary ovarian cancer (r = -0.64, p = 0.002). The correlation was even stronger when only high grade serous tumors were counted (r = -0.868, p < 0.001) (Table S2). In metastatic tumors Ang-2 PP was strongly correlated to the corresponding mRNA levels (r = 0.752, p = 0.005) (Table S2). Tie-1 and Tie-2 qRT-PCR values correlated strongly to each other (r = 0.919, p < 0.001), but did not have statistically significant correlations to the respective immunohistochemical expression. In metastases, mRNA levels of Ang-2 correlated strongly with Tie-2 mRNA (r= 0.61, p= 0.016).

Modified sentences, page 14, rows 262-69.

Likewise in the caption:

S2 Table.

Correlation of qRT-PCR results with immunohistochemical staining. Ang-2 IRS and PP were correlated with relative mRNA levels as measured by qRT-PCR. There were 22 primary tumor samples and 15 metastatic samples in each group.

Modified caption, page 28, rows 613-15.

P14 lines 276-277: delete “the” before “ovarian” and “greater”

Deleted ”the”, page 14, rows 273-74.

P15 line 280: delete “the” before “resistance”

Deleted ”the”, page 15, row 277.

P15 line 297: delete “the” before “ovarian”

Deleted ”the”, page 15, row 294.

P16 line 324: delete “the” before “significance”

Deleted ”the”, page 16, row 321. 

P19 line 354: amend “expressed widely” to “widely expressed”

Amended the expression, page 19, row 351.

P21 line 390: add comma after “in addition”

Added comma, page 21, row 387.

P21 line 398: amend to “Expression of Ang-2 and Tie-2”

Amended, page 21, row 395.

P22 line 414-415: delete “the” before “omental”; amend “exfoliating” to “exfoliation”; add “the” before “omental surface”

Modified, page 22, rows 411-12.

P22 lines 425-427: delete “the” before “lower” and “larger residual tumour”. Also every instance of “the” could be deleted in the sentence beginning “The expression of Tie-1”.

Modified, page 22, rows 422-24.

P23 line 434: amend to “Ang-2 specific andtibody and VEGF blocker aflibercept more strongly reduced tumour growth”

Modified, page 22, row 431.

P23 line 453: delete “the” before “OS”.

Deleted ”the”, page 23, row 450. 

Response to Reviewer 4

We thank Reviewer for the valuable comment. Your comment or question is indicated in bold followed by our response. All changes are also highlighted in the “Revised manuscript with track changes” and corrected in the unmarked version of revised manuscript without track changes.

4. Have the authors made all data underlying the findings in their manuscript fully available?

 Reviewer #4: No

We think, that all necessery data underlying the findings of the manuscript is written in the manuscript. If additional information concerning the data is desired, it is fully available from the authors upon reasonable request.

---

## [Decision Letter · Decision Letter 4]

23 Sep 2020

PONE-D-19-33270R4

HIGH EXPRESSION OF TIE-2 PREDICTS POOR PROGNOSIS IN PRIMARY HIGH GRADE SEROUS OVARIAN CANCER

PLOS ONE

Dear Dr. Sopo,

Thank you for submitting your manuscript to PLOS ONE. After careful consideration, we feel that it has merit but does not fully meet PLOS ONE’s publication criteria as it currently stands. Therefore, we invite you to submit a revised version of the manuscript that addresses the points raised during the review process.

We look forward to receiving your revised manuscript.

Kind regards,

Elizabeth Christie

Academic Editor

PLOS ONE

Reviewers' comments:

Reviewer's Responses to Questions

**Comments to the Author**

1. If the authors have adequately addressed your comments raised in a previous round of review and you feel that this manuscript is now acceptable for publication, you may indicate that here to bypass the “Comments to the Author” section, enter your conflict of interest statement in the “Confidential to Editor” section, and submit your "Accept" recommendation.

Reviewer #3: (No Response)

Reviewer #4: (No Response)

2. Is the manuscript technically sound, and do the data support the conclusions?

Reviewer #3: Yes

Reviewer #4: Yes

3. Has the statistical analysis been performed appropriately and rigorously? 

Reviewer #3: Yes

Reviewer #4: Yes

4. Have the authors made all data underlying the findings in their manuscript fully available?

Reviewer #3: Yes

Reviewer #4: No

5. Is the manuscript presented in an intelligible fashion and written in standard English?

Reviewer #3: Yes

Reviewer #4: Yes

6. Review Comments to the Author

Reviewer #3: All of my previous comments have been satisfactorily addressed. However there are still some problems with Table S3.

Original Q7: a) some of the data in Table S3 still seems to be misaligned. Should all of the p values cited for the Stage analyses be aligned with the III-IV rows rather than the I-II rows? Similarly, should the p values related to Recurrence all be aligned with the Yes rows?

b) Thankyou for your explanation of the reporting of p values for serous vs total tumours in Table. S3. Please explain this more clearly in the legend, e.g. “* p value reported for serous tumours when only comparison for this subgroup was statistically significant”. However, if this is the case, why are the p values 0.938 and 0.605 corresponding to Tie1 staining with residual tumour <1cm also marked with *? Do these p values represent the whole study population, or serous tumours only? To avoid confusion, please present the all p values (significant or not) for the whole study population in Table S3 to ensure consistency throughout the table. For the instances where the p value was only statistically significant for serous tumours, include this value with * and in brackets after the value for the whole study population; e.g. 0.###(0.008*).

Reviewer #4: Please carefully review the PLOS Data Policy https://journals.plos.org/plosone/s/data-availability

The following points extracted from the Policy are particularly relevant:

"PLOS journals require authors to make all data necessary to replicate their study’s findings publicly available without restriction at the time of publication. When specific legal or ethical restrictions prohibit public sharing of a data set, authors must indicate how others may obtain access to the data."

"Authors must share the “minimal data set” for their submission. PLOS defines the minimal data set to consist of the data required to replicate all study findings reported in the article, as well as related metadata and methods... For example, authors should submit the following data: The values behind the means, standard deviations and other measures reported;

The values used to build graphs..."

"Stating ‘data available on request from the author’ is not sufficient."

"PLOS recognizes that, in some instances, authors may not be able to make their underlying data set publicly available for legal or ethical reasons. This data policy does not overrule local regulations, legislation or ethical frameworks. Where these frameworks prevent or limit data release, authors must make these limitations clear in the Data Availability Statement at the time of submission... For studies involving human research participant data or other sensitive data, we encourage authors to share de-identified or anonymized data. However, when data cannot be publicly shared, we allow authors to make their data sets available upon request. If there are ethical or legal restrictions on sharing a sensitive data set, authors should provide the following information within their Data Availability Statement upon submission: Explain the restrictions in detail (e.g., data contain potentially identifying or sensitive patient information); Provide contact information for a data access committee, ethics committee, or other institutional body to which data requests may be sent."

The authors have stated in their Data Availability Statement that "All relevant data are within the manuscript and its Supporting Information files." This is not the case, as all data necessary to replicate the various analyses, tables and figures is currently not in the Supporting Information files. Please refer to the PLOS Data policy.

7. PLOS authors have the option to publish the peer review history of their article (what does this mean?). If published, this will include your full peer review and any attached files.

Reviewer #3: No

Reviewer #4: No

---

## [Author Response · Author response to Decision Letter 4]

5 Oct 2020

A point-by-point Response letter to the Reviewers

Response to Reviewer 3

We thank Reviewer for the valuable comments. Each of your comment or question is indicated in bold followed by our response. All changes are also highlighted in the “Revised manuscript with track changes” and corrected in the unmarked version of revised manuscript without track changes.

Reviewer #3: All of my previous comments have been satisfactorily addressed. However there are still some problems with Table S3.

Original Q7: a) some of the data in Table S3 still seems to be misaligned. Should all of the p values cited for the Stage analyses be aligned with the III-IV rows rather than the I-II rows? Similarly, should the p values related to Recurrence all be aligned with the Yes rows?

P-values for the stage analysis are aligned with the III-IV rows and p-values of the recurrence are aligned with the Yes rows in Table S3 of Supporting information. 

b) Thank you for your explanation of the reporting of p values for serous vs total tumours in Table. S3. Please explain this more clearly in the legend, e.g. “* p value reported for serous tumours when only comparison for this subgroup was statistically significant”. However, if this is the case, why are the p values 0.938 and 0.605 corresponding to Tie1 staining with residual tumour <1cm also marked with *? Do these p values represent the whole study population, or serous tumours only? To avoid confusion, please present the all p values (significant or not) for the whole study population in Table S3 to ensure consistency throughout the table. For the instances where the p value was only statistically significant for serous tumours, include this value with * and in brackets after the value for the whole study population; e.g. 0.###(0.008*).

”p value reported for serous tumours when only comparison for this subgroup was statistically significant” is now explained in the legend of Table S3, page 28, rows 618-19 and under the Table S3 in Supporting information. 

P values 0.938 and 0.605 corresponding to Tie1 residual tumor <1cm represent serous tumors when comparing the <1cm group to 0 group, as the only significance was found in serous group (>1cm vs 0). Now we have changed the p values of those particular lines to represent the values of whole study population in Table S3. 

We have now presented all p values for the whole study population and when the p value of only serous tumors was significant, it is included in brackets after the value of whole study population in Table S3. 

Response to Reviewer 4

We thank Reviewer for the valuable comment. Your comment or question is indicated in bold followed by our response. All changes are also highlighted in the “Revised manuscript with track changes” and corrected in the unmarked version of revised manuscript without track changes.

Reviewer #4: Please carefully review the PLOS Data Policy https://journals.plos.org/plosone/s/data-availability

The following points extracted from the Policy are particularly relevant:

"PLOS journals require authors to make all data necessary to replicate their study’s findings publicly available without restriction at the time of publication. When specific legal or ethical restrictions prohibit public sharing of a data set, authors must indicate how others may obtain access to the data."

"Authors must share the “minimal data set” for their submission. PLOS defines the minimal data set to consist of the data required to replicate all study findings reported in the article, as well as related metadata and methods... For example, authors should submit the following data: The values behind the means, standard deviations and other measures reported;

The values used to build graphs..."

"Stating ‘data available on request from the author’ is not sufficient."

We have now included the ”minimal data set” in excel format as Supporting information in the submission according to the PLOS Data Policy.

---

## [Decision Letter · Decision Letter 5]

16 Oct 2020

HIGH EXPRESSION OF TIE-2 PREDICTS POOR PROGNOSIS IN PRIMARY HIGH GRADE SEROUS OVARIAN CANCER

PONE-D-19-33270R5

Dear Dr. Sopo,

We’re pleased to inform you that your manuscript has been judged scientifically suitable for publication and will be formally accepted for publication once it meets all outstanding technical requirements.

Kind regards,

Elizabeth Christie

Academic Editor

PLOS ONE

Additional Editor Comments (optional):

Reviewers' comments:

Reviewer's Responses to Questions

**Comments to the Author**

1. If the authors have adequately addressed your comments raised in a previous round of review and you feel that this manuscript is now acceptable for publication, you may indicate that here to bypass the “Comments to the Author” section, enter your conflict of interest statement in the “Confidential to Editor” section, and submit your "Accept" recommendation.

Reviewer #3: All comments have been addressed

Reviewer #4: All comments have been addressed

2. Is the manuscript technically sound, and do the data support the conclusions?

Reviewer #3: Yes

Reviewer #4: Yes

3. Has the statistical analysis been performed appropriately and rigorously? 

Reviewer #3: Yes

Reviewer #4: Yes

4. Have the authors made all data underlying the findings in their manuscript fully available?

Reviewer #3: Yes

Reviewer #4: Yes

5. Is the manuscript presented in an intelligible fashion and written in standard English?

Reviewer #3: Yes

Reviewer #4: Yes

6. Review Comments to the Author

Reviewer #3: (No Response)

Reviewer #4: (No Response)

7. PLOS authors have the option to publish the peer review history of their article (what does this mean?). If published, this will include your full peer review and any attached files.

Reviewer #3: No

Reviewer #4: No

---

## [Editor Report · Acceptance letter]

21 Oct 2020

PONE-D-19-33270R5 

High expression of Tie-2 predicts poor prognosis in primary high grade serous ovarian cancer 

Dear Dr. Sopo:

I'm pleased to inform you that your manuscript has been deemed suitable for publication in PLOS ONE. Congratulations! Your manuscript is now with our production department. 

Kind regards, 

on behalf of

Dr. Elizabeth Christie 

Academic Editor

PLOS ONE